# SKETCHED GAUSSIAN MECHANISM ON MATRIX FOR PRIVATE FEDERATED LORA

## ABSTRACT

Low-Rank Adaptation (LoRA), which modifies frozen pre-trained parameters via the product of two trainable low-rank factors, has been widely adopted for communication-efficient fine-tuning of language models, including extensions to federated learning (FL). Nevertheless, two challenges arise at scale: (i) for very large models, the adapter factors can remain high-dimensional, leading to nontrivial communication costs between clients and the server; and (ii) transmitting local adapters between clients and the server risks privacy leakage. Incorporating differential privacy (DP) by additive mechanisms, e.g., the Gaussian mechanism (GM), often leads to substantial noise amplification, particularly in algorithms that must perturb both low-rank components.

In this paper, we propose the Sketched Gaussian Mechanism on Matrix (SGMM), which couples random sketching with the Gaussian mechanism at the matrix level. Using tools from Rényi differential privacy (RDP), we provide a unified analysis of SGMM's privacy guarantee and show that, for a fixed privacy level, the required noise magnitude scales as $1/\sqrt{b}$ for sketch dimension $b$. Consequently, for moderate $b$, SGMM attains the same privacy with markedly less noise than GM. We instantiate SGMM within federated LoRA algorithms, including FFA-LoRA and FlexLoRA, where sketching further reduces adapter dimensionality and, in turn, the noise needed to meet a given privacy target, addressing both communication overhead and noise amplification. Experiments demonstrate that, at matched privacy budgets, SGMM-based federated LoRA is at least competitive with and in some settings outperforms non-sketched private baselines.

## 1 INTRODUCTION

Recent years have witnessed remarkable advances in large language models (LLMs) (Touvron et al., 2023; Achiam et al., 2023; Zhang et al., 2022a; Zeng et al., 2022). Despite their strong general capabilities, modern LLMs often contain trillions of parameters, making full-parameter fine-tuning computationally prohibitive. As a remedy, parameter-efficient fine-tuning (PEFT) methods (Ding et al., 2023), in particular Low-Rank Adaptation (LoRA) (Hu et al., 2022), have been widely adopted across downstream tasks. LoRA augments a frozen pre-trained weight matrix with a product of two trainable low-rank factors, thereby updating only a small fraction of the total parameters while keeping the base weights fixed. Relative to full fine-tuning, it typically achieves comparable or even superior performance with substantially improved computational efficiency.

This parameter efficiency aligns naturally with federated learning (FL). First, because FL clients are resource-constrained, LoRA reduces local storage and computation burdens (Hu et al., 2022). Second, communication reduction is key to fine-tuning in FL (Malaviya et al., 2023); in principle, federated LoRA communicates only the low-rank factors, yielding markedly lower communication costs. Third, as data remain on-device with no direct sharing, federated fine-tuning can provide stronger privacy protection than centralized fine-tuning. However, integrating LoRA into FL still poses challenges: (i) despite transmitting only the adapter matrices, their non low-rank dimensions (e.g., $d$ and $k$ for $d \times r$ and $r \times k$ matrices) can remain large for significantly large base models, leading to substantial communication overhead when exchanging adapters between server and clients; (ii) although FL is often viewed as privacy-preserving by virtue of on-device training, it does not by itself provide rigorous guarantees such as differential privacy (DP) (Dwork et al., 2006) and is vulnerable to inference attacks that can leak client information during training (Nasr et al., 2019; Pustozerova & Mayer, 2020; Xie et al., 2019; Zhao et al., 2020; Zhu et al., 2019). These issues motivate communication-reduced and privacy-aware federated LoRA designs.

Regarding communication cost, a natural approach is to reduce the non-rank dimensions of the adapter matrices via compression. Numerous efforts in standard federated learning have pursued this direction (Kairouz et al., 2021; Li et al., 2014), including sparsification (Lin et al., 2017; Wang et al., 2018; Barnes et al., 2020), quantization (Alistarh et al., 2017; Liu et al., 2023; Reisizadeh et al., 2020), and sketching (Song et al., 2023; Rothchild et al., 2020; Ivkin et al., 2019). Among these, sketching methods stand out for their simplicity, which facilitates integration with existing FL pipelines. As an unbiased compressor, sketching avoids the bias corrections and associated memory overhead often required by sparsification via error-feedback mechanisms (Seide et al., 2014; Stich et al., 2018). Moreover, being linear, sketching approximately preserves geometric structure after compression (Dasgupta & Gupta, 2003), in contrast to quantization, which can distort inner products and potentially slow convergence (Alistarh et al., 2017).

Regarding privacy, two DP notions are commonly used in FL design: sample-level and client-level privacy. Client-level privacy is stricter, requiring that outputs remain statistically indistinguishable even when an entire client's dataset is changed, rather than only the inclusion/exclusion of a single example as in sample-level privacy (Zhang et al., 2022b). Many FL methods (Geyer et al., 2017; Wang et al., 2020; Triastcyn & Faltings, 2019; Truex et al., 2020) target client-level privacy by adapting the centralized DP recipe, i.e., clipping and then adding Gaussian noise to the clipped quantities (Abadi et al., 2016; Fang et al., 2023), to the federated setting. In the federated LoRA context, however, a direct application of this mechanism necessitates perturbing both adapter matrices, thereby doubling the injected noises. Moreover, the LoRA update involves a matrix product $BA$, which further intensifies noise amplification. Prior work has sought to mitigate this problem (Sun et al., 2024; Lee et al., 2025; Wen et al., 2025). In particular, FFA-LoRA (Sun et al., 2024) freezes one adapter at random initialization so that only the remaining adapter is perturbed during training.

In this paper, we introduce the Sketched Gaussian Mechanism on Matrix (SGMM), which combines an isometric Gaussian sketching transform (Song et al., 2023) with the classical Gaussian mechanism, thereby achieving both dimensionality reduction and privacy protection. Using analysis from Rényi differential privacy (RDP) (Mironov, 2017), we derive a tight upper bound on the overall privacy level $\epsilon$ of SGMM. Specifically, holding all other hyperparameters fixed, $\epsilon = O\left(\frac{1}{\sqrt{b}\sigma_g^2}\right)$, where $b$ is the sketch dimension and $\sigma_g^2$ is the variance of the injected Gaussian noise. This establishes an explicit dependence of $\epsilon$ on both $b$ and $\sigma_g^2$, and implies that, for sufficiently large $b$, SGMM attains strictly stronger privacy guarantees than the standard GM (Gaussian mechanism), demonstrating that the sketching step itself contributes inherent privacy. While SGM has been recently developed for vectors (Li et al., 2025), permitting a vectorization-based application to matrices, our work shows substantial computational benefits of SGMM, along with a unique analysis for privacy of SGMM depending on suitable covariance matrices and matrix norms. In fact, the recent analysis on SGM of vectors can be seen as a special case of our new SGMM analysis applied to a matrix with one column.

We further integrate SGMM into federated LoRA algorithms, including FFA-LoRA (Sun et al., 2024) and FlexLoRA (Bai et al., 2024), yielding SGMM-FFA-LoRA and SGMM-FlexLoRA respectively, and prove that these integrations satisfy client-level privacy guarantees with subsampling, post-processing, and composition properties of differential privacy (Dwork et al., 2014). In addition, we empirically validate our approach on deep learning models. Across these tasks, the SGMM variants of federated LoRA require strictly less Gaussian noise than their GM counterparts at the same privacy level, has less communication overhead due to sketching of factor matrices, and consistently achieving comparable or superior accuracy.

The remainder of the paper is organized as follows. Section 2 introduces SGMM and establishes its privacy guarantees. Section 3 presents the integration of SGMM with FFA-LoRA and FlexLoRA and derives the corresponding client-level privacy results. Section 4 reports experimental evaluations, comparing SGMM-based federated LoRA algorithms with their counterparts using the standard Gaussian mechanism. Section 5 concludes and outlines future directions. Owing to space constraints, our discussion on related works appears in Appendix A.

## 2 PRIVACY GUARANTEE OF SGM ON MATRIX (SGMM)

In this section, we study sketching-based privacy mechanisms for matrix statistics, presenting two different mechanisms and providing rigorous analyses of their privacy guarantees. We study the privacy guarantee subject to the rigorous privacy guarantees of Differential Privacy (DP)Dwork et al. (2006), whose formal definition is given below.

**Definition 2.1.** A randomized algorithm $\mathcal{M}$ is $(\epsilon, \delta)$-differentially private if for any pair of datasets $D, D'$ differ in exactly one data point and for all event $\mathcal{Y}$ in the output range of $\mathcal{M}$, we have

$$\mathbb{P}\left\{\mathcal{M}(D) \in \mathcal{Y}\right\} \leq e^{\epsilon} \mathbb{P}\left\{\mathcal{M}(D') \in \mathcal{Y}\right\} + \delta$$

where the probability is taken over the randomness of $\mathcal{M}$.

## 2.1 Privacy based on SGM with Vectorization (SGMV) of Matrix

Because matrix-valued statistics are more challenging to handle directly, and most differential-privacy mechanisms are formulated for vector-valued quantities (e.g., gradient updates), a natural reduction is to apply the vectorization operator $\text{vec}(\cdot)$ to map a matrix to a vector and then deploy a vector-level mechanism, e.g., standard GM. The recently proposed Sketched Gaussian Mechanism (SGM) of Li et al. (2025) provides a privacy analysis with sketching for vector-valued statistics; we briefly review SGM below and adopt it for matrix-valued statistics via vectorization.

**Definition 2.2** (Sketched Gaussian Mechanism (SGM)). For any statistic $\theta(D) \in \mathbb{R}^d$ of the dataset $D$, the Sketched Gaussian Mechanism outputs $\mathcal{SG}(\theta; R, \xi) = R\theta + \xi$, in which $R \in \mathbb{R}^{b \times d}$ is a Gaussian sketching matrix with each entry sampled i.i.d. from $\mathcal{N}(0, \frac{1}{\sqrt{b}})$ and $\xi \in \mathbb{R}^b$ follows the Gaussian distribution $\mathcal{N}\left(0, \sigma_g^2 \mathbb{I}_b\right)$.

Implied by the proof of Theorem 2.2 in Li et al. (2025), SGM satisfies the following privacy guarantee.

**Theorem 2.1** (Li et al. (2025)). Assume $\|\theta\|_2 \leq \tau$. There exists constants $c_1$ and $c_2$ such that for any $\epsilon_p \leq c_1$, SGM is $(\epsilon_p, \delta_p)$-differentially private for any $\delta_p > 0$ if we choose

$$\sigma_g^2 \geq \frac{c_2 \tau^2 \sqrt{\ln(1/\delta_p)}}{\sqrt{b}\epsilon_p} \ . \tag{1}$$

By incorporating with vectorization, we can define the Sketched Gaussian Mechanism with Vectorization (SGMV) on matrix $\gamma(D) \in \mathbb{R}^{m \times r}$ and obtain the privacy guarantee.

**Definition 2.3** (Sketched Gaussian Mechanism with Vectorization (SGMV)). For any matrix statistic $\gamma(D) \in \mathbb{R}^{m \times r}$ of the dataset $D$, the Sketched Gaussian Mechanism with Vectorization (SGMV) outputs $\mathcal{SGV}(\gamma; R, \xi) = R\text{vec}(\gamma) + \xi$, in which $R \in \mathbb{R}^{h \times mr}$ is a Gaussian sketching matrix with each entry sampled i.i.d. from $\mathcal{N}(0, \frac{1}{\sqrt{h}})$ and $\xi \in \mathbb{R}^h$ follows the Gaussian distribution $\mathcal{N}(0, \sigma_g^2 \mathbb{I}_h)$.

**Theorem 2.2.** Assume $\|\gamma(D)\|_{\text{F}} \leq \tau$. There exists constants $c_1$ and $c_2$ such that for any $\epsilon_p \leq c_1$, SGMV is $(\epsilon_p, \delta_p)$-differentially privacy for any $\delta_p > 0$ if we choose

$$\sigma_g^2 \geq \frac{c_2 \tau^2 \sqrt{\ln(1/\delta_p)}}{\sqrt{h}\epsilon_p} \ .$$

**Beyond Vectorization.** Although vectorization provides a simple route to apply sketching-based privacy to matrices, it has a critical limitation: SGMV yields an $h$-dimensional vector and thereby discards the matrix's intrinsic two-dimensional structure (e.g., row/column couplings and low-rank geometry) that is crucial in many applications, including LoRA. Furthermore, if we set $h = br$ so the sketched vector can be reshaped into a $b \times r$ matrix, according to Definition 2.3 and Definition 2.4, SGMV forms $R\text{vec}(\gamma)$ with $R \in \mathbb{R}^{br \times mr}$ and $\text{vec}(\gamma) \in \mathbb{R}^{mr}$, which costs $\Theta(br \cdot mr) = \Theta(bmr^2)$. In contrast, SGMM computes $R\gamma$ with $R \in \mathbb{R}^{b \times m}$ and $\gamma \in \mathbb{R}^{m \times r}$, which costs $\Theta(bmr)$, yielding that SGMV incurs an $O(r)$-fold higher computation. Thus, sketching the vectorized matrix is computationally more demanding. Such considerations motivate the development and analysis of mechanisms that operate natively on matrices and preserve their structural properties with lower computational requirements.

## 2.2 Privacy based on SGM on Matrix (SGMM)

Similar to SGM in Definition 2.2, now we define the Sketched Gaussian Mechanism on Matrix (SGMM).

**Definition 2.4** (Sketched Gaussian Mechanism on Matrix (SGMM)). For any matrix statistic $\gamma(D) \in \mathbb{R}^{m \times r}$ of the dataset $D$, the Sketched Gaussian Mechanism on Matrix outputs $\mathcal{SGM}(\gamma; R, \xi) = R\gamma + \xi$, in which $R \in \mathbb{R}^{b \times m}$ is a Gaussian sketching matrix with each entry sampled i.i.d. from $\mathcal{N}(0, \frac{1}{\sqrt{b}})$ and $\xi \in \mathbb{R}^{b \times r}$ is a noise matrix with independent entries sampled from $\mathcal{N}(0, \sigma_g^2)$.

Next, we state the privacy guarantee of SGMM:

**Theorem 2.3.** Assume $\|\gamma(D)\|_{\text{F}} \leq \tau$. There exists constants $c_3$ and $c_4$ such that for for any $\epsilon_p \leq c_3$, SGMM is $(\epsilon_p, \delta_p)$-differentially privacy for any $\delta_p > 0$ if we choose

$$\sigma_g^2 \geq \frac{c_4 \sqrt{r} \tau^2 \sqrt{\ln(1/\delta_p)}}{\sqrt{b}\epsilon_p} \ .$$

**Remark 2.1.** While Theorem 2.3 analyzes SGMM in the matrix setting, it inherits a fundamental feature of SGM's result on vectors in Theorem 2.1: for a prescribed privacy level $\epsilon_p$, the required Gaussian variance $\sigma_g^2$ decreases monotonically with the sketch dimension $b$. Consequently, increasing $b$ within a moderate range strictly lowers the variance required by the mechanism. Moreover, the classical GM requires $\sigma_g^2 \geq \frac{C\tau^2 \ln(1.25/\delta_p)}{\epsilon_p^2}$ (Dwork et al., 2014, Theorem 3.22), so SGMM attains the same privacy level with smaller noise whenever the sketch dimension satisfies $b \geq \Omega\left(\frac{r\epsilon_p^2}{\ln(1/\delta_p)}\right)$. $\quad\square$

**Remark 2.2.** Comparing SGMV and SGMM, beyond SGMM's advantage of preserving the matrix's two-dimensional structure, there is a clear privacy–compute trade-off. To compare on equal footing, assume both mechanisms produce outputs with the same number of entries, i.e., set $h = br$.

On the computational side, as discussed in "Beyond Vectorization", SGMM's computation cost is lower than SGMV's by a factor of $r$. On the privacy side, for the same injected noise level, Theorem 2.2 yields that the privacy level of SGMV is on the order $\Theta(1/\sqrt{br})$, whereas Theorem 2.3 shows that SGMM attains a privacy level of order $\Theta(\sqrt{r}/\sqrt{b})$. Hence, under the same noise magnitude, $\epsilon_p$ increases by a factor of $r$, implying less privacy. Taken together, these observations formalize the trade-off between SGMV and SGMM in terms of computational complexity versus privacy level. $\quad\square$

We refer readers to Appendix B.1 for a detailed proof, and we provide a high-level sketch here. Our analysis leverages tools from the Rényi Differential Privacy (RDP) framework (Mironov, 2017).

**Definition 2.5** (Rényi divergence (Rényi, 1961)). For two probability distributions $P$ and $Q$ defined over $\mathcal{R}$, the Rényi divergence of order $\alpha > 1$ is $D_\alpha(P\|Q) \triangleq \frac{1}{\alpha-1} \ln \mathbb{E}_{x \sim Q}\left(\frac{P(x)}{Q(x)}\right)^\alpha$.

By definition of SGMM, $\mathcal{SGM}(\gamma; R, \xi) \in \mathbb{R}^{b \times r}$ consists of $b$ columns, each distributed as $\mathcal{N}(0, \frac{\gamma^\top \gamma}{b} + \sigma_g^2 \mathbb{I})$. This observation yields the following result:

**Lemma 2.4.** For any constant $c \geq 0$, assume $\Sigma_D^c = \gamma(D)^\top \gamma(D) + c^2\mathbb{I}$, $\Sigma_{D'}^c = \gamma(D')^\top \gamma(D') + c^2\mathbb{I}$. Then the Rényi divergence between $\mathcal{SGM}(\gamma; R, \xi)$ for neighboring datasets $D$ and $D'$ is

$$D_\alpha\left(\mathcal{SGM}(\gamma(D); R, \xi)\|\mathcal{SGM}(\gamma(D'); R, \xi)\right) = \frac{b}{2(\alpha-1)} \ln \frac{\left(\det \Sigma_D^{\sqrt{b}\sigma_g}\right)^{1-\alpha}\left(\det \Sigma_{D'}^{\sqrt{b}\sigma_g}\right)^\alpha}{\det\left((1-\alpha)\Sigma_D^{\sqrt{b}\sigma_g} + \alpha\Sigma_{D'}^{\sqrt{b}\sigma_g}\right)}.$$

To make the expression amenable to analysis, we define the matrix $M_{D,D'}^c := (\Sigma_D^c)^{-\frac{1}{2}} \Sigma_{D'}^c (\Sigma_D^c)^{-\frac{1}{2}}$.

**Lemma 2.5.** The Rényi divergence between $\mathcal{SGM}(\gamma; R, \xi)$ for neighboring datasets $D$ and $D'$ is

$$D_\alpha\left(\mathcal{SGM}(\gamma(D); R, \xi)\|\mathcal{SGM}(\gamma(D'); R, \xi)\right) = \frac{b}{2(\alpha-1)} \sum_{i=1}^r f_\alpha(\lambda_i),$$

where $f_\alpha(x) = \alpha \ln x - \ln(1 - \alpha + \alpha x)$, and $\lambda_1 \geq \cdots \geq \lambda_r$ are the eigenvalues of $M_{D,D'}^{\sqrt{b}\sigma_g}$.

Since $f_\alpha$ is monotonically increasing when $x \geq 1$ and decreasing when $x \leq 1$, it follows that $f_\alpha(\lambda_i)$ attains its maximum over $\{\lambda_1, ..., \lambda_r\}$ at either $\lambda_1$ or $\lambda_r$. Hence,

$$D_\alpha\left(\mathcal{SGM}(\gamma(D); R, \xi)\|\mathcal{SGM}(\gamma(D'); R, \xi)\right) = \frac{b}{2(\alpha-1)} \sum_{i=1}^r f_\alpha(\lambda_i) \leq \frac{br}{2(\alpha-1)} \max\left\{f_\alpha(\lambda_1), f_\alpha(\lambda_r)\right\}.$$

Thus, we obtain a bound on the Rényi divergence that depends only on the largest and smallest eigenvalues of $M_{D,D'}^{\sqrt{b}\sigma_g}$. Now we introduce the upper and lower ratio sensitivity of the statistic $\gamma$ analogous to the classical sensitivity measure in standard GM (Dwork et al., 2014).

**Definition 2.6** (Upper/Lower Ratio Sensitivity). For any constant $c \geq 0$, define the ratio sensitivity of the statistic $\gamma$ as

$$\overline{\text{rsens}_c}(\gamma) = \sup_{D,D'} \lambda_{\max}\left(M_{D,D'}^c\right), \qquad \underline{\text{rsens}_c}(\gamma) = \inf_{D,D'} \lambda_{\min}\left(M_{D,D'}^c\right).$$

where the supremum and infimum are over all neighboring datasets $D, D'$.

We next bound the upper and lower ratio sensitivities, by invoking Weyl's inequality, stated below.

**Theorem 2.6** (Weyl's inequality Weyl (1912)). Let $A, B$ be Hermitian on inner product space $V$ with dimension $n$, with spectrum ordered in descending order $\lambda_1 \geq \cdots \lambda_n$. Then we have

$$\lambda_{i+j-1}(A + B) \leq \lambda_i(A) + \lambda_j(B) \leq \lambda_{i+j-n}(A + B).$$

According to our definition of $M_{D,D'}^{\sqrt{b}\sigma_g}$, it is a symmetric matrix and can be written as

$$M_{D,D'}^{\sqrt{b}\sigma_g} = \left(\Sigma_D^{\sqrt{b}\sigma_g}\right)^{-\frac{1}{2}} \Sigma_{D'}^{\sqrt{b}\sigma_g} \left(\Sigma_D^{\sqrt{b}\sigma_g}\right)^{-\frac{1}{2}} = \mathbb{I} + \left(\Sigma_D^{\sqrt{b}\sigma_g}\right)^{-\frac{1}{2}} \left(\Sigma_{D'}^{\sqrt{b}\sigma_g} - \Sigma_D^{\sqrt{b}\sigma_g}\right) \left(\Sigma_D^{\sqrt{b}\sigma_g}\right)^{-\frac{1}{2}}$$

$$= \mathbb{I} + \left(\Sigma_D^{\sqrt{b}\sigma_g}\right)^{-\frac{1}{2}} \left(\gamma(D')^{\top}\gamma(D') - \gamma(D)^{\top}\gamma(D)\right) \left(\Sigma_D^{\sqrt{b}\sigma_g}\right)^{-\frac{1}{2}}.$$

Applying Theorem 2.6 with $A = \mathbb{I}, B = \mathbb{I} + \left(\Sigma_D^{\sqrt{b}\sigma_g}\right)^{-\frac{1}{2}} \left(\gamma(D')^{\top}\gamma(D') - \gamma(D)^{\top}\gamma(D)\right) \left(\Sigma_D^{\sqrt{b}\sigma_g}\right)^{-\frac{1}{2}}$,
this gives

$$1 - \frac{2\tau^2}{b\sigma_g^2} \le \lambda_{\min}\left(M_{D,D'}^{\sqrt{b}\sigma_g}\right) \le \lambda_{\max}\left(M_{D,D'}^{\sqrt{b}\sigma_g}\right) \le 1 + \frac{2\tau^2}{b\sigma_g^2}.$$

Since this inequality holds for all $D, D'$, we can further obtain

$$1 - \frac{2\tau^2}{b\sigma_g^2} \le \underline{\mathrm{rsens}}_{\sqrt{b}\sigma_g}(\gamma) \le \overline{\mathrm{rsens}}_{\sqrt{b}\sigma_g}(\gamma) \le 1 + \frac{2\tau^2}{b\sigma_g^2}.$$

With the monotonicity of $f_\alpha$, we can obtain the following bound on the Rényi divergence.

**Lemma 2.7.** For any pair of neighboring datasets $D, D'$,

$$D_\alpha\left(\mathcal{SGM}(\gamma(D); R, \xi)\|\mathcal{SGM}(\gamma(D'); R, \xi)\right) \le \frac{br}{2(\alpha-1)} \max\left\{f_\alpha\left(1 + \frac{2\tau^2}{b\sigma_g^2}\right), f_\alpha\left(1 - \frac{2\tau^2}{b\sigma_g^2}\right)\right\} \le \frac{r\alpha^2\tau^4}{(\alpha-1)b\sigma_g^4}.$$

We now proceed to analyze the privacy of SGMM under Rényi Differential Privacy (RDP).

**Definition 2.7** (($\alpha, \epsilon$)-RDP Mironov (2017)). A randomized mechanism $f : \mathcal{D} \to \mathcal{R}$ is said to have $\epsilon$-Rényi differential privacy of order $\alpha$, or ($\alpha, \epsilon$)-RDP for short, if for any adjacent $D, D' \in \mathcal{D}$, it holds that $D_\alpha\left(f(D)\|f(D')\right) \le \epsilon$.

Furthermore, RDP can be transformed into the ($\epsilon, \delta$)-differential privacy guarantee.

**Lemma 2.8** (Relationship with ($\epsilon, \delta$)-DP Mironov (2017)). If f is an ($\alpha, \epsilon$)-RDP mechanism, it also satisfies $\left(\epsilon + \frac{\ln 1/\delta}{\alpha-1}, \delta\right)$-differential privacy for any $0 < \delta < 1$.

Invoking Lemma 2.7 and Lemma 2.8 yields the RDP and ($\epsilon, \delta$)-DP results for SGMM.

**Lemma 2.9.** SGMM on the statistic $\gamma$ is $\left(\alpha, \frac{r\alpha^2\tau^4}{(\alpha-1)b\sigma_g^4}\right)$-RDP, therefore $\left(\frac{r\alpha^2\tau^4}{(\alpha-1)b\sigma_g^4} + \frac{\ln(1/\delta)}{\alpha-1}, \delta\right)$-DP. By optimizing the Rényi order $\alpha$ as in Lemma 2.4, we recover the optimal ($\epsilon, \delta$)-DP guarantee for SGMM given in Theorem 2.3.

## 3 APPLICATION OF SGMM IN FEDERATED LoRA ALGORITHMS

In this section, we integrate SGMV and SGMM into different federated LoRA algorithms, specifically FFA-LoRA (Sun et al., 2024) and FlexLoRA (Bai et al., 2024), to achieve communication efficiency together with differential-privacy guarantees without incurring significant noise amplification.
We consider a federated learning setup with $C$ clients indexed by $c \in [C]$. Client $c$ possesses a local dataset $\mathcal{D}_c = \{(x_{i,c}, y_{i,c})\}_{i=1}^{n_c}$ of size $n_c$, and defines its empirical risk $\mathcal{L}_c\left(A, B; W_0\right) = \frac{1}{n_c}\sum_{i=1}^{n_c} \ell\left(W_0 + BA, (x_{i,c}, y_{i,c})\right)$, where $W_0 \in \mathbb{R}^{d\times k}$ denotes the frozen base weights, $A \in \mathbb{R}^{r\times k}, B \in \mathbb{R}^{d\times r}$ are the two adapter matrices, and $\ell$ is the per-example loss. The term $W_0 + BA$ is interpreted layerwise, i.e., at each LoRA-instrumented layer $l \in [L]$, the effective weight is $W_{0,l} + B_l A_l$. The global objective is to minimize the average empirical loss across clients $\mathcal{L} := \frac{1}{C}\sum_{c=1}^{C}\mathcal{L}_c$.

### 3.1 SGMV/SGMM-FFA-LoRA

Algorithm 1 presents SGMV/SGMM-FFA-LoRA, i.e., the incorporation of SGMV/SGMM within the FFA-LoRA framework (Sun et al., 2024). Among the two adapter matrices, we fix the right matrix $A$ at its initialization $A_0$ and update only the left matrix $B$ during federated training. At each communication round $t$, the server samples $N$ clients uniformly at random from the $C$ available clients. Every selected client $c$ performs $K$ steps of local stochastic gradient descent (SGD) on $B$ using its local dataset, obtaining $B_c^{t,K}$. The client then applies Frobenius-norm clipping at threshold $\tau_B$ to produce $\hat{B}_c^{t,K}$. Two options follow:

1. Apply SGMV to $\hat{B}_c^{t,K}$ to obtain a sketched vector $\tilde{B}_c^{t,K}$ and transmit it to the server. The server aggregates the received sketches to form the global update $B^{t+1}$ and broadcasts it. Each client de-sketches the sketch to recover a vector in $\mathbb{R}^{dr}$ and then applies devec (grouping every $d$ coordinates into a column) to reshape it into $B_c^{t,0} \in \mathbb{R}^{d\times r}$ for the local initialization of the next round.

---

**Algorithm 1** SGMV-FFA-LoRA/SGMM-FFA-LoRA

---

**Hyperparameters:** local learning rate $\eta_{\text{local}}$, number of local epochs/steps $K$, number of rounds $T$, client fraction $q = \frac{N}{C}$, LoRA rank $r$.

**Inputs:** client datasets $D_c = \{(x_{i,c}, y_{i,c})\}_{i=1}^{n_c}$ for $c \in [C]$; frozen base weights $W_0$; **fixed** LoRA right matrices $A_0 \in \mathbb{R}^{r \times k}$ (random init, then frozen); initial left matrices $B^0 \in \mathbb{R}^{d \times r}$ with $B^0 = 0$; loss

$$\mathcal{L}_c (B; W_0, A_0) = \frac{1}{n_c} \sum_{i=1}^{n_c} \ell \left( W_0 + BA_0, (x_{i,c}, y_{i,c}) \right).$$

    **Initialize** $B^0 \leftarrow 0$; sample and **fix** $A_0$; freeze $W_0$.
    **for** $t = 1, \ldots, T - 1$ **do**
        Take a subset $\mathcal{C}_t \subseteq [C]$ of $N$ clients with sampling probability $q = \frac{N}{C}$.
        **On Client Nodes:**
        **for** $c \in \mathcal{C}_t$ **do**
            **Local init:** $B_c^{t,0} \leftarrow \text{devec} \left( R_B^{t-1 \top} B^{t-1} \right)$   (use **fixed** $A_0$; freeze $W$)
            $\left( \text{\textbf{Local init:} } B_c^{t,0} \leftarrow R_B^{t \top} B^t \right)$
            **Local update:**
            **for** $k = 1, \ldots, K$ **do**
                Sample a mini-batch $(x, y) \sim D_c$
                Compute gradient: $g_{c,t,k}^B = \nabla_B \ell \left( W_0 + B_c^{t,k-1} A_0, (x, y) \right)$
                Update left factors only: $B_c^{t,k} \leftarrow B_c^{t,k-1} - \eta_{\text{local}} g_{c,t,k}^B$
            **end for**
            **Clipping:** $\hat{B}_c^{t,K} = \text{clip} \left( B_c^{t,K}, \tau_B \right) = B_c^{t,K} \cdot \min \left\{ 1, \frac{\tau_B}{\|B_c^{t,K}\|_{\text{F}}} \right\}$
            **Apply SGMV:** $\tilde{B}_c^{t,K} = \mathcal{SGV} \left( \hat{B}_c^{t,K}; R_B^t, \mathbf{z}_{c,B}^t \right)$
            $\left( \text{\textbf{Apply SGMM:}} \tilde{B}_c^{t,K} = \mathcal{SGM} \left( \hat{B}_c^{t,K}; R_B^t, \mathbf{z}_{c,B}^t \right) \right)$
            **Send parameters:** transmit final left factors $\tilde{B}_c^{t,K}$ to the server
        **end for**
        **On Server Node:**
        **Aggregation:** $B^{t+1} \leftarrow \frac{1}{N} \sum_{c \in \mathcal{C}_t} \tilde{B}_c^{t,K}$
        Broadcast $B^{t+1}$ (and the fixed $A_0$ if needed) to all clients
    **end for**
**Outputs:** final LoRA adapters $(A_0^T, B^T)$.

---

    2. Apply SGMM to $\hat{B}_c^{t,K}$ to obtain a sketched matrix $\tilde{B}_c^{t,K}$ and transmit it to the server. The subsequent aggregation and broadcast proceed analnously, except that the matrix structure is preserved throughout and no devec operation is required.

**Remark 3.1.** In the algorithm, the rationale for applying $R_B^{t \top}$ for de-sketching, followed by devec to reshape back into $\mathbb{R}^{d \times r}$ relies on the isometry of the sketching matrix $R_B^t$. Take the SGMV variant of the algorithm as an example. According to our definitions of the sketching matrix $R_B^t$ and the random Gaussian noise $\mathbf{z}_{c,B}^{t-1}$ in Definition 2.3, we have $\mathbb{E} \left[ R_B^{t-1 \top} R_B^{t-1} \right] = \mathbb{I}$, $\mathbb{E} \left[ R_B^{t-1} \right] = 0$, and $\mathbb{E} \left[ \mathbf{z}_{c,B}^{t-1} \right] = 0$. Thus, for any matrix $M$, writing $(M)_j$ for its $j$-th column, and $(R_B^{t-1 \top} B^{t-1})^{(j)}$ for its $j$-th group of $d$ coordinates, the algorithm implies

$$\mathbb{E} \left[ (R_B^{t-1 \top} B^{t-1})^{(j)} \right] = \mathbb{E} \left[ \left( R_B^{t-1 \top} \left( \frac{1}{N} \sum_{c \in \mathcal{C}_{t-1}} \tilde{B}_c^{t-1,K} \right) \right)_j \right]$$

$$= \mathbb{E} \left[ R_B^{t-1 \top} R_B^{t-1} \left( \frac{1}{N} \sum_{c \in \mathcal{C}_{t-1}} \hat{B}_c^{t-1,K} \right)_j \right] + \mathbb{E} \left[ R_B^{t-1 \top} \left( \frac{1}{N} \sum_{c \in \mathcal{C}_{t-1}} \mathbf{z}_{c,B}^{t-1} \right)_j \right] = \left( \frac{1}{N} \sum_{c \in \mathcal{C}_{t-1}} \hat{B}_c^{t-1,K} \right)_j.$$

  Therefore, after the devec step, $\mathbb{E} \left[ B_c^{t,0} \right] = \frac{1}{N} \sum_{c \in \mathcal{C}_{t-1}} \hat{B}_c^{t-1,K}$, which coincides with the aggregated, clipped adapter matrices $B$ from the participating clients.

Applying Theorem 2.2 and Theorem 2.3 in conjunction with the subsampling lemma, post-processing invariance, and the sequential composition theorem of Dwork et al. (2014) yields the following client-level privacy guarantees for Algorithm 1.

---

**Algorithm 2** SGMM-FlexLoRA

---

**Hyperparameters:** local learning rate $\eta_{\text{local}}$, number of local steps/epochs $K$, number of rounds $T$, client fraction $q = \frac{N}{C}$, per-client ranks $\{r_c\}_{c \in [C]}$, global rank $r$, LoRA scale $s$.

**Inputs:** local datasets $D_c = \{(x_{i,c}, y_{i,c})\}_{i=1}^{n_c}$ for clients $c \in [C]$; frozen base weights $W_0$; initial factors $A^0 \in \mathbb{R}^{r \times k}$ and $B^0 \in \mathbb{R}^{d \times r}$; loss

$$\mathcal{L}_c(A, B; W_0) = \frac{1}{n_c} \sum_{i=1}^{n_c} \ell(W_0 + BA, (x_{i,c}, y_{i,c})).$$

   **Initialize** $(A^0, B^0)$; freeze $W_0$

   **for** $t = 0, \ldots, T - 1$ **do**

      Take a subset $\mathcal{C}_t$ of $N$ clients with sampling probability $q = \frac{N}{C}$

      **On Client Nodes:**

      **for** $c \in \mathcal{C}_t$ **do**

         **Local initialization:** $A_c^{t,0} \leftarrow A^t R_A^t$,    $B_c^{t,0} \leftarrow R_B^{t\top} B^t$   (freeze $W$)

         **Local update (train $A$ and $B$):**

         **for** $k = 1, \ldots, K$ **do**

            Sample a mini-batch $(x, y) \sim D_c$.

            Compute gradients:

            $g_{c,t,k}^A = \nabla_A \ell(W_0 + B_c^{t,k-1} A_c^{t,k-1}, (x, y)), g_{c,t,k}^B = \nabla_B \ell(W_0 + B_c^{t,k-1} A_c^{t,k-1}, (x, y))$

            Update both factors: $A_c^{t,k} \leftarrow A_c^{t,k-1} - \eta_{\text{local}} g_{c,t,k}^A$,    $B_c^{t,k} \leftarrow B_c^{t,k-1} - \eta_{\text{local}} g_{c,t,k}^B$

         **end for**

         **Clipping:** $\hat{A}_c^{t,K} = \text{clip}(A_c^{t,K}, \tau_A)$, $\hat{B}_c^{t,K} = \text{clip}(B_c^{t,K}, \tau_B)$

         **Apply SGMM:** $\tilde{A}_c^{t,K} = \mathcal{SGM}(\hat{A}_c^{t,K}; R_A^t, \mathbf{z}_{c,A}^t)$, $\tilde{B}_c^{t,K} = \mathcal{SGM}(\hat{B}_c^{t,K}; R_B^t, \mathbf{z}_{c,B}^t)$

         **Send parameters:** transmit final adapters $(\tilde{A}_c^{t,K}, \tilde{B}_c^{t,K})$ to the server

      **end for**

      **On Server Node:**

      **Aggregate full LoRA weights:** $W^{t+1} \leftarrow \frac{1}{N} \sum_{c \in \mathcal{C}_t} \tilde{B}_c^{t,K} \tilde{A}_c^{t,K}$

      **Single SVD (once per round):**   $(U^{t+1}, \Sigma^{t+1}, V^{t+1\top}) \leftarrow \text{SVD}(W^{t+1})$

      **Set global max-rank factors:**

$$B^{t+1} \leftarrow U^{t+1}[:, 1:r]\,\Sigma^{t+1}[1:r, 1:r], \quad A^{t+1} \leftarrow V^{t+1\top}[1:r, :].$$

$$(B^{t+1} \leftarrow U^{t+1}[:, 1:r]\,\Sigma^{t+1}[1:r, 1:r]^{\frac{1}{2}}, \quad A^{t+1} \leftarrow \Sigma^{t+1}[1:r, 1:r]^{\frac{1}{2}} V^{t+1\top}[1:r, :])$$

      Broadcast updated adapters $(A^{t+1}, B^{t+1})$ to all clients.

   **end for**

**Outputs:** global LoRA update $B^T A^T$.

---

**Theorem 3.1** (Client-level Privacy of SGMV-FFA-LoRA). *Denote $h_B$ the sketching dimension of SGMV on $B$, and $\sigma_{g,B}^2$ the variance of $\mathbf{z}_{c,B}^t$. There exists constants $c_5$ and $c_6$ such that given the sampling probability $q = \frac{N}{C}$ and the number of communication rounds $T$, for any $\epsilon_p \leq c_5 q \sqrt{T}$, SGMV-FFA-LoRA is $(\epsilon_p, \delta_p)$-differentially privacy for any $\delta_p > 0$ if we choose*

$$\sigma_{g,B}^2 \geq \frac{c_6 q \tau_B^2 \sqrt{T} \ln(2qT/\delta_p)}{\sqrt{h_B} \epsilon_p}.$$

**Theorem 3.2** (Client-level Privacy of SGMM-FFA-LoRA). *Denote $b_B$ the sketching dimension of SGMM on $B$, and $\sigma_{g,B}^2$ the variance of $\mathbf{z}_{c,B}^t$. There exists constants $c_7$ and $c_8$ such that given the sampling probability $q = \frac{N}{C}$ and the number of communication rounds $T$, for any $\epsilon_p \leq c_7 q \sqrt{T}$, SGMM-FFA-LoRA is $(\epsilon_p, \delta_p)$-differentially privacy for any $\delta_p > 0$ if we choose*

$$\sigma_{g,B}^2 \geq \frac{c_8 q \tau_B^2 \sqrt{rT} \ln(2qT/\delta_p)}{\sqrt{b_B} \epsilon_p}.$$

## 3.2 SGMM-FLEXLORA

We formalize SGMM-FlexLoRA in Algorithm 2 by embedding SGMM within the FlexLoRA algorithm (Bai et al., 2024). In each communication round $t$, the server samples a subset of $N$ clients at random from the $C$ total clients. Every selected client $c$ performs $K$ steps of local SGD on both adapter matrices $A$ and $B$ using its local dataset, reaching $A_c^{t,K}$ and $B_c^{t,K}$. Next, the client clips with

respect to the Frobenius norm at level $\tau$ to obtain $\hat{A}_c^{t,K}$ and $\hat{B}_c^{t,K}$, applies SGMM to each to produce $\tilde{A}_c^{t,K}$ and $\tilde{B}_c^{t,K}$, and transmits both to the server. The server aggregates the products $\tilde{B}_c^{t,K}\tilde{A}_c^{t,K}$ to form $W^{t+1}$, then computes an SVD of $W^{t+1}$ to obtain $U^{t+1}$, $\Sigma^{t+1}$, and $V^{t+1\top}$. Using this decomposition, the server redistributes the LoRA factors into two adapters $A^{t+1}$ and $B^{t+1}$ with rank at most $r$ by selecting the top-$r$ singular vectors and corresponding singular values, and broadcasts them back to the clients. Each client then inverts these two matrices back to the original dimensions, yielding $A_c^{t+1,0} \in \mathbb{R}^{r \times k}$ and $B_c^{t+1,0} \in \mathbb{R}^{d \times r}$, as the initialization of the next round.

**Remark 3.2.** Unlike FFA-LoRA, SGMV cannot be incorporated into FlexLoRA. The key distinction is that FlexLoRA multiplies the two adapters at the server to form the full LoRA weights. If SGMV were applied to both $\hat{A}_c^{t,K}$ and $\hat{B}_c^{t,K}$, sending $\tilde{A}_c^{t,K} \in \mathbb{R}^{h_A}$ and $\tilde{B}_c^{t,K} \in \mathbb{R}^{h_B}$ produced via sketching matrices $R_A^t \in \mathbb{R}^{h_A \times rk}$ and $R_B^t \in \mathbb{R}^{h_B \times dr}$ respectively, the resulting weight matrix at the server would be $W^{t+1} \in \mathbb{R}^{h_B \times h_A}$. After SVD decomposition and redistribution, this yields $A^{t+1} \in \mathbb{R}^{r \times h_A}$ and $B^{t+1} \in \mathbb{R}^{h_B \times r}$. Inverting with the corresponding sketching matrices then produces $A_c^{t,0} \in \mathbb{R}^{r \times rk}$ and $B_c^{t,0} \in \mathbb{R}^{dr \times r}$, which have incorrect dimensions. By contrast, in FFA-LoRA, where $A$ is fixed and no such multiplication is performed, SGMV remains feasible.

In a similar manner as Algorithm 1, we apply Theorem 2.3 together with subsampling, post-processing, and the composition theorems in Dwork et al. (2014) to derive the following client-level privacy result for Algorithm 2.

**Theorem 3.3** (Client-level Privacy of SGMM-FlexLoRA). Denote $b_A$ and $b_B$ the sketching dimensions of SGMV on $A$ and $B$, and $\sigma_{g,A}^2$ and $\sigma_{g,B}^2$ the variance of $\mathbf{z}_{c,A}^t$ and $\mathbf{z}_{c,B}^t$. There exists constants $c_7$ and $c_8$ such that given the sampling probability $q = \frac{N}{C}$ and the number of communication rounds $T$, for any $\epsilon_p \le c_7 q\sqrt{T}$, SGMM-FlexLoRA is $(2\epsilon_p, 2\delta_p)$-differentially privacy for any $\delta_p > 0$ if we choose

$$\sigma_{g,A}^2 \ge \frac{c_8 q \tau_A^2 \sqrt{rT} \ln(2qT/\delta_p)}{\sqrt{b_A}\epsilon_p}, \quad \sigma_{g,B}^2 \ge \frac{c_8 q \tau_B^2 \sqrt{rT} \ln(2qT/\delta_p)}{\sqrt{b_B}\epsilon_p}.$$

**Remark 3.3.** In Section 3, all algorithms and theorems assume fixed clipping thresholds uniform across rounds and clients, i.e., $\tau_A$ for the adapter matrix $A$ and $\tau_B$ for the adapter matrix $B$. In fact, the results can be extended directly to adaptive clipping, provided the corresponding noise requirements are adjusted. For example, assume that at communication round $t \in [T]$ and client $c \in \mathcal{C}_t$, we clip $A_c^{t,K}$ at the clipping threshold $\tau_{A,c}^t$, then the Gaussian noise at that round $t$ and client $c$ must satisfy $\sigma_{g,A,c}^{t}{}^2 \ge \frac{c_8 q \tau_{A,c}^{t}{}^2 \sqrt{rT} \ln(2qT/\delta_p)}{\sqrt{b_A}\epsilon_p}$. A similar condition holds for the matrix $B$.

**Remark 3.4.** In the SGMM-based algorithms and privacy analysis, we clip the adapter matrices $A$ and $B$ with respect to the Frobenius norm. However, according to our proof in Appendix B, we can see that analogous results can be obtained when clipping is performed under other matrix norms, e.g., the spectral norm or the $2, \infty$ norm. For instance, if spectral-norm clipping is used, the same noise-variance requirement as shown in Theorem 3.2 and Theorem 3.3 applies. If instead the $2, \infty$ norm is used, the requirements of the variances become $\sigma_{g,A}^2 \ge \frac{c_8 q \tau_A^2 \sqrt{r^3 T} \ln(2qT/\delta_p)}{\sqrt{b_A}\epsilon_p}$ for $A$ and $\sigma_{g,B}^2 \ge \frac{c_8 q \tau_B^2 \sqrt{r^3 T} \ln(2qT/\delta_p)}{\sqrt{b_B}\epsilon_p}$ for $B$, which differ by a factor of $r$.

**Remark 3.5.** While throughout Section 3, the algorithms are instantiated with SGD for both local and server-side updates and the privacy analysis is conducted under this choice, by the post-processing invariance of differential privacy (Dwork et al., 2014), all stated privacy guarantees continue to hold if SGD is replaced with other optimizers on the client and/or server, including adaptive methods such as Adam (Kingma & Ba, 2014), AdaGrad (Duchi et al., 2011), and Yogi (Zaheer et al., 2018).

## 4 EXPERIMENTS

We empirically evaluate all proposed algorithms—SGMM-FFA-LoRA, SGMV-FFA-LoRA, and SGMM-FlexLoRA—on large-model fine-tuning. For each algorithm, we compare against its counterpart using the standard Gaussian mechanism under the same privacy level. The results follow a detailed description of the experimental setup.

**Datasets and Network Structure.** Regarding the dataset, we use CIFAR-100, which contains 50K training images and 10K test images across 100 classes, consisting of $32 \times 32$ color images spanning diverse object categories. Regarding the model, we fine-tune a ViT-based architecture (Vaswani et al., 2017) using LoRA adapters with 2.4M parameters, approximately 2.4% of the overall 86M model size.

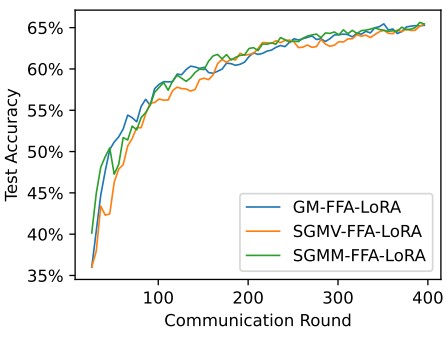 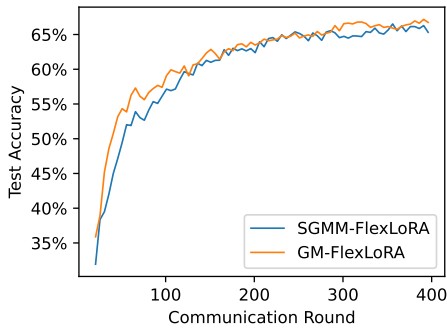

(a) Private FFA-LoRA, $\epsilon_p = 1.70$      (b) Private FlexLoRA, $\epsilon_p = 1.70$

Figure 1: (a) Comparison of different variants of private FFA-LoRA algorithms under the same privacy $\epsilon_p = 1.70$, including SGMV-FFA-LoRA with noise multiplier $0.73$, SGMM-FFA-LoRA with noise multiplier $1.45$, and FFA-LoRA with the traditional GM (GM-FFA-LoRA) with noise multiplier $1.50$. (b) Comparison of different variants of private FlexLoRA algorithms under the same privacy $\epsilon_p = 1.70$, including SGMM-FlexLoRA with noise multiplier $1.68$, and FlexLoRA with the traditional GM (GM-FlexLoRA) with noise multiplier $1.99$. The X-axis is the number of communication rounds $T$, and the Y-axis is the test accuracy.

**Parameter Setting.** We deploy $C = 625$ clients and sample $N = 4$ clients uniformly at random per communication round. Each selected client performs $K = 10$ local updates on mini-batches of size 8. The clipping threshold is set to the 50% quantile of the norm distribution at the current round. The non–low-rank dimensions are $d = k = 768$, the low-rank dimension is $r = 4$, and the number of layers is $L = 12$. With an approximate 20% compression rate, we use the sketching dimension $b = 150$ for SGMM-FFA-LoRA and SGMM-FlexLoRA, and $br = 600$ for SGMV-FFA-LoRA. We run all experiments for $T = 400$ communication rounds.

**Privacy Level and Noise calculation.** For privacy, we fix $\delta_p = 10^{-5}$ and target $\epsilon_p = 1.70$. Under this privacy level, the required noise multiplier $\sigma_g/\tau$ are as follows: with the traditional Gaussian mechanism, $1.50$ for FFA-LoRA and $1.99$ for FlexLoRA; with SGMV-FFA-LoRA, $0.73$; and with the SGMM variants, $1.45$ for SGMM-FFA-LoRA and $1.68$ for SGMM-FlexLoRA.

**Experimental Results.** We report our experimental results in Figure 1. Each experiment is repeated 5 times, and the average curves are shown in the plots. Figure 1a depicts test accuracy for three private FFA-LoRA variants: SGMM-FFA-LoRA, SGMV-FFA-LoRA, and GM-FFA-LoRA, and Figure 1b depicts test accuracy for two private FlexLoRA variants: SGMM-FlexLoRA and GM-FlexLoRA. All models are evaluated at the same privacy level $\epsilon_p = 1.70$. Across both settings, the sketched methods consistently equal or exceed the performance of the non-sketched baselines, indicating that sketching is effective in a differentially private federated LoRA framework. Specifically, in the FFA-LoRA case, SGMM-FFA-LoRA and SGMV-FFA-LoRA both slightly outperform the GM-FFA-LoRA baseline. This performance gain stems from the fact that the sketched mechanisms require less Gaussian noise and fewer noise dimensions to meet the same privacy target, so that the reduced injection of noise can counteract the performance losses that sketching sometimes introduces.

## 5 CONCLUSION

In summary, we proposed the Sketched Gaussian Mechanism on Matrix (SGMM), which couples an isometric matrix sketch with the classical Gaussian mechanism. Using tools from Rényi differential privacy, we established that the privacy loss satisfies $\epsilon_p = O\left(1/\sqrt{b}\sigma_g^2\right)$, thereby exhibiting inherent privacy amplification due to sketching. We further integrated SGMM into federated LoRA algorithms, specifically FFA-LoRA and FlexLoRA, and proved client-level privacy guarantees for these methods by invoking subsampling, post-processing, and composition theorems of differential privacy. It is also confirmed empirically that, for both FFA-LoRA and FlexLoRA, the sketched private variants achieve a fixed privacy budget with strictly less noise and consistently meet or exceed the accuracy of their unsketched counterparts. As a direction for future work, our analysis presently focuses on isotropic Gaussian sketching matrices; establishing comparable privacy and convergence guarantees for broader families of sketches remains an open problem.

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

## A    RELATED WORK

**Parameter efficient fine-tuning.** As large language models (LLMs) continue to grow, full fine-tuning becomes increasingly costly and often impractical. Parameter-efficient fine-tuning (PEFT) addresses this by introducing a small set of trainable parameters $\Theta$ while freezing most pre-trained parameters $\Phi$; the task-specific increment $\Delta\Phi$ is thereby encoded in a much lower-dimensional $\Delta\Theta$. Representative PEFT approaches include adapter layers inserted into each network block (Houlsby et al., 2019), and prefix-tuning (Li & Liang, 2021) or prompt-tuning (Lester et al., 2021), which augment the model by concatenating trainable dimensions to input or hidden representations. Another line of work, including LoRA (Hu et al., 2022) and RGP (Yu et al., 2021), uses low-rank matrices to approximate or re-parameterize pre-trained weights. LoRA, in particular, has become a leading PEFT method: it tunes fewer than $5\%$ of the parameters required by full fine-tuning yet attains comparable performance across diverse downstream tasks, and even can be adapted to new tasks (Yang et al., 2024; Agiza et al., 2024). This parameter efficiency aligns naturally with federated learning (FL), and numerous prior works have explored such integration (Sun et al., 2024; Lee et al., 2025; Wen et al., 2025).

**Communication-Efficient Distributed Learning.** The high cost of transmitting model updates between clients and a central server has motivated extensive work on communication efficiency in distributed and federated learning. A widely used strategy, FedAvg (McMahan et al., 2017), reduces communication frequency by permitting multiple local updates per round before synchronization. A complementary direction compresses updates prior to transmission, thereby lowering the per-round communication load. Such compression methods broadly include sparsification (Aji & Heafield, 2017; Wang et al., 2018; Lin et al., 2017), quantization (Suresh et al., 2017; Alistarh et al., 2017; Wen et al., 2017), low-rank factorization (Vargaftik et al., 2021; Mohtashami et al., 2022; Vogels et al., 2019), sketching (Rothchild et al., 2020; Song et al., 2023; Jiang et al., 2018), and sparse subnetwork training (Isik et al., 2022; Li et al., 2020; 2021). While some techniques, notably certain quantization schemes (Alistarh et al., 2017), are inherently unbiased, many introduce bias and thus require additional corrections to improve convergence (Lin et al., 2017; Rothchild et al., 2020). Linearity is another key property, as it helps preserve geometric structure after compression (Dasgupta & Gupta, 2003). Within this landscape, sketching stands out for its simplicity: it is a linear, unbiased transformation that enables computation in a reduced dimensional space before reconstruction via desketching.

**Privacy in Federated Learning.** Differential Privacy (DP) (Dwork et al., 2006) provides a rigorous standard for privacy guarantees in machine learning. In centralized training, the canonical DP procedure clips each stochastic gradient and adds Gaussian noise to the clipped gradient (Abadi et al., 2016). The clipping operation is pivotal, since the required noise variance depends directly on the chosen clipping threshold (Dwork et al., 2014). Clipping-based mechanisms are also widely employed in federated learning, though differing requirements lead to distinct clipping strategies. For sample-level privacy, one approach clips and injects noise to every local update (Truex et al., 2019), while often with a noticeable performance drop. For client-level privacy, two common strategies are: (i) clip the local updates and add random noises before transmission to the server (Geyer et al., 2017; Wang et al., 2020; Triastcyn & Faltings, 2019); or (ii) clip the local models prior to transmission and perturb the bounded parameters (Wei et al., 2020; Truex et al., 2020).

**Sketching.** For decades, well before the rise of deep learning in the 2010s, sketching has been a core tool across diverse applications (Cormode & Muthukrishnan, 2005; Greenwald & Khanna, 2001; Kane & Nelson, 2014). It is widely used for low-rank approximation (Tropp et al., 2017), graph sparsification (Ahn et al., 2012), and least-squares regression (Dobriban & Liu, 2019). More recently, sketching has been adopted in distributed and federated learning to compress model updates and thereby improve communication efficiency (Jiang et al., 2018; Ivkin et al., 2019; Rothchild et al., 2020; Song et al., 2023; Haddadpour et al., 2020; Shrivastava et al., 2024). Sketching-based pipelines have also been seamlessly integrated with secure aggregation and differential privacy mechanisms (Chen et al., 2022; Song et al., 2023; Melis et al., 2015; Zhu et al., 2020; Bassily et al., 2017). However, most existing approaches analyze privacy only for the randomness injected after sketching and do not quantify any privacy amplification contributed by the sketching transformation itself. Li et al. (2019) and Li et al. (2025) are among the few works that study this intrinsic privacy effect of sketching.

# B   PRIVACY ANALYSIS

In this section, we will provide the proof of Theorem 2.3, Theorem 3.2, and Theorem 3.3 on the privacy guarantees of the SGMM mechanism, Algorithm 1 and Algorithm 2 respectively.

## B.1   PROOF OF THEOREM 2.3

We will establish our analysis in the framework of Mironov (2017). We will restate the concepts and propositions here.

**Definition B.1** (Rényi divergence). For two probability distributions $P$ and $Q$ defined over $\mathcal{R}$, the Rényi divergence of order $\alpha > 1$ is

$$D_\alpha\left(P\|Q\right) \triangleq \frac{1}{\alpha-1} \ln \mathbb{E}_{x\sim Q} \left(\frac{P(x)}{Q(x)}\right)^\alpha.$$

**Definition B.2** ($(\alpha, \epsilon)$-RDP). A randomized mechanism $f : \mathcal{D} \to \mathcal{R}$ is said to have $\epsilon$-Rényi differential privacy of order $\alpha$, or $(\alpha, \epsilon)$-RDP for short, if for any adjacent $D, D' \in \mathcal{D}$, it holds that

$$D_\alpha\left(f(D)\|f(D')\right) \leq \epsilon.$$

**Lemma B.1** (Lemma 2.4). For any constant $c \geq 0$, assume $\Sigma_D^c = \gamma(D)^\top\gamma(D) + c^2\mathbb{I}$, $\Sigma_{D'}^c = \gamma(D')^\top\gamma(D') + c^2\mathbb{I}$. Then the Rényi divergence between $\mathcal{SGM}(\gamma; R, \xi)$ for neighboring datasets $D$ and $D'$ is

$$D_\alpha\left(\mathcal{SGM}(\gamma(D); R, \xi)\|\mathcal{SGM}(\gamma(D'); R, \xi)\right) = \frac{b}{2(\alpha-1)} \ln \frac{\left(\det \Sigma_D^{\sqrt{b}\sigma_g}\right)^{1-\alpha} \left(\det \Sigma_{D'}^{\sqrt{b}\sigma_g}\right)^\alpha}{\det\left((1-\alpha)\Sigma_D^{\sqrt{b}\sigma_g} + \alpha\Sigma_{D'}^{\sqrt{b}\sigma_g}\right)}.$$

*Proof.* According to the definition of SGMM, we have

$$(\mathcal{SGM}(\gamma(D); R, \xi))_j \sim \mathcal{N}\left(0, \frac{\gamma(D)^\top\gamma(D)}{b} + \sigma_g^2\mathbb{I}\right);$$

$$(\mathcal{SGM}(\gamma(D'); R, \xi))_j \sim \mathcal{N}\left(0, \frac{\gamma(D')^\top\gamma(D')}{b} + \sigma_g^2\mathbb{I}\right).$$

Following the definition of Rényi divergence in Definition B.1, we can get that

$$
\begin{aligned}
&D_\alpha \left(\mathcal{SGM}(\gamma(D); R, \xi) \| \mathcal{SGM}(\gamma(D'); R, \xi)\right)\\
=&b \cdot D_\alpha \left((\mathcal{SGM}(\gamma(D); R, \xi))_j \| (\mathcal{SGM}(\gamma(D'); R, \xi))_j\right)\\
=&bD_\alpha \left(\mathcal{N}\left(0, \frac{\gamma(D)^\top \gamma(D)}{b} + \sigma_g^2 \mathbb{I}\right) \middle\| \mathcal{N}\left(0, \frac{\gamma(D')^\top \gamma(D')}{b} + \sigma_g^2 \mathbb{I}\right)\right)\\
=&\frac{b}{2(\alpha - 1)} \ln \frac{\left(\det\left(\frac{\gamma(D)^\top \gamma(D)}{b} + \sigma_g^2 \mathbb{I}\right)\right)^{1-\alpha} \left(\det\left(\frac{\gamma(D')^\top \gamma(D')}{b} + \sigma_g^2 \mathbb{I}\right)\right)^\alpha}{\det\left((1-\alpha)\left(\frac{\gamma(D)^\top \gamma(D)}{b} + \sigma_g^2 \mathbb{I}\right) + \alpha\left(\frac{\gamma(D')^\top \gamma(D')}{b} + \sigma_g^2 \mathbb{I}\right)\right)}\\
=&\frac{b}{2(\alpha - 1)} \ln \frac{\left(\det\left(\frac{\Sigma_D^{\sqrt{b}\sigma_g}}{b}\right)\right)^{1-\alpha} \left(\det\left(\frac{\Sigma_{D'}^{\sqrt{b}\sigma_g}}{b}\right)\right)^\alpha}{\det\left(\frac{(1-\alpha)\Sigma_D^{\sqrt{b}\sigma_g} + \alpha\Sigma_{D'}^{\sqrt{b}\sigma_g}}{b}\right)}\\
=&\frac{b}{2(\alpha - 1)} \ln \frac{\left(\det \Sigma_D^{\sqrt{b}\sigma_g}\right)^{1-\alpha} \left(\det \Sigma_{D'}^{\sqrt{b}\sigma_g}\right)^\alpha}{\det\left((1-\alpha)\Sigma_D^{\sqrt{b}\sigma_g} + \alpha\Sigma_{D'}^{\sqrt{b}\sigma_g}\right)}.
\end{aligned}
$$

$\square$

**Lemma B.2** (Lemma 2.5). For any constant $c \geq 0$, define $M_{D,D'}^c = (\Sigma_D^c)^{-\frac{1}{2}} \Sigma_{D'}^c (\Sigma_D^c)^{-\frac{1}{2}}$. Then the Rényi divergence between $\mathcal{SGM}(\gamma; R, \xi)$ for neighboring datasets $D$ and $D'$ is

$$
D_\alpha \left(\mathcal{SGM}(\gamma(D); R, \xi) \| \mathcal{SGM}(\gamma(D'); R, \xi)\right) = \frac{b}{2(\alpha - 1)} \sum_{i=1}^r f_\alpha(\lambda_i),
$$

in which $f_\alpha = \alpha \ln x - \ln(1 - \alpha + \alpha x)$, and $\lambda_1 \geq \cdots \geq \lambda_r$ are the eigenvalues of $M_{D,D'}^{\sqrt{b}\sigma_g}$.

*Proof.* We first prove the following general lemma.

**Lemma B.3.** Assume $\Sigma, \Sigma' \in \mathbb{R}^{r \times r}$ are two positive definite matrices. Denote $M = \Sigma^{-\frac{1}{2}} \Sigma' \Sigma^{-\frac{1}{2}}$, and $\lambda_1 \geq \cdots \geq \lambda_r$ are the eigenvalues of $M$. Then we have

$$
\ln \frac{(\det \Sigma)^{1-\alpha} (\det \Sigma')^\alpha}{\det((1-\alpha)\Sigma + \alpha\Sigma')} = \sum_{i=1}^r f_\alpha(\lambda_i)
$$

in which $f_\alpha = \alpha \ln x - \ln(1 - \alpha + \alpha x)$.

*Proof.* Since $M = \Sigma^{-\frac{1}{2}} \Sigma' \Sigma^{-\frac{1}{2}}$, we have

$$
\Sigma' = \Sigma^{\frac{1}{2}} \left(\Sigma^{-\frac{1}{2}} \Sigma' \Sigma^{\frac{1}{2}}\right) \Sigma^{\frac{1}{2}} = \Sigma^{\frac{1}{2}} M \Sigma^{\frac{1}{2}},
$$

$$
\begin{aligned}
(1-\alpha)\Sigma + \alpha\Sigma' &= \Sigma^{\frac{1}{2}}(1-\alpha)\mathbb{I}_r \Sigma^{\frac{1}{2}} + \Sigma^{\frac{1}{2}}\alpha M \Sigma^{\frac{1}{2}}\\
&= \Sigma^{\frac{1}{2}}((1-\alpha)\mathbb{I}_r + \alpha M)\Sigma^{\frac{1}{2}}
\end{aligned}
$$

By the multiplicativity of determinants,

$$
\det \Sigma' = \det\left(\Sigma^{\frac{1}{2}} M \Sigma^{\frac{1}{2}}\right) = \det \Sigma^{\frac{1}{2}} \det M \det \Sigma^{\frac{1}{2}} = \det \Sigma \det M,
$$

$$
\begin{aligned}
\det((1-\alpha)\Sigma + \alpha\Sigma') &= \det\left(\Sigma^{\frac{1}{2}}((1-\alpha)\mathbb{I}_r + \alpha M)\Sigma^{\frac{1}{2}}\right) = \det \Sigma^{\frac{1}{2}} \det((1-\alpha)\mathbb{I}_r + \alpha M) \det \Sigma^{\frac{1}{2}}\\
&= \det \Sigma \det((1-\alpha)\mathbb{I}_r + \alpha M).
\end{aligned}
$$

Therefore,

$$
\ln \frac{(\det \Sigma)^{1-\alpha} (\det \Sigma')^\alpha}{\det((1-\alpha)\Sigma + \alpha\Sigma')}
$$

$$= (1-\alpha)\ln\det\Sigma + \alpha\ln\det\Sigma' - \ln\det\left((1-\alpha)\Sigma + \alpha\Sigma\right)$$
$$= (1-\alpha)\ln\det\Sigma + \alpha\ln\left(\det\Sigma\det M\right) - \ln\left(\det\Sigma\det\left((1-\alpha)\mathbb{I}_r + \alpha M\right)\right)$$
$$= (1-\alpha)\ln\det\Sigma + \alpha\ln\det\Sigma + \alpha\ln\det M - \ln\det\Sigma - \ln\det\left((1-\alpha)\mathbb{I}_r + \alpha M\right)$$
$$= \alpha\ln\det M - \ln\det\left((1-\alpha)\mathbb{I}_r + \alpha M\right)$$

There exists orthonormal matrix $P$ s.t. $M = P\mathrm{diag}\left(\lambda_1, ..., \lambda_r\right)P^\top$, so

$$(1-\alpha)\mathbb{I}_r + \alpha M = (1-\alpha)PP^\top + \alpha P\mathrm{diag}\left(\lambda_1, ..., \lambda_r\right)P^\top$$
$$= P\mathrm{diag}\left((1-\alpha), ..., (1-\alpha)\right)P^\top + P\mathrm{diag}\left(\lambda_1, ..., \lambda_r\right)P^\top$$
$$= P\mathrm{diag}\left(1-\alpha+\alpha\lambda_1, ..., 1-\alpha+\alpha\lambda_r\right)P^\top$$

By the multiplicativity of determinants,

$$\det M = \det\left(P\mathrm{diag}\left(\lambda_1, ..., \lambda_r\right)P^\top\right)$$
$$\det P\det P^\top\det\mathrm{diag}\left(\lambda_1, ..., \lambda_r\right)$$
$$= \prod_{i=1}^r \lambda_i;$$
$$\det\left((1-\alpha)\mathbb{I}_r + \alpha M\right) = \det\left(P\mathrm{diag}\left(1-\alpha+\alpha\lambda_1, ..., 1-\alpha+\alpha\lambda_r\right)P^\top\right)$$
$$= \det P\det P^\top\det\mathrm{diag}\left(1-\alpha+\alpha\lambda_1, ..., 1-\alpha+\alpha\lambda_r\right)$$
$$= \prod_{i=1}^r\left(1-\alpha+\alpha\lambda_i\right).$$

So we can get that

$$\alpha\ln\det M - \ln\det\left((1-\alpha)\mathbb{I}_r + \alpha M\right)$$
$$= \alpha\ln\prod_{i=1}^r\lambda_i - \ln\prod_{i=1}^r\left(1-\alpha+\alpha\lambda_i\right)$$
$$= \alpha\sum_{i=1}^r\ln\lambda_i - \sum_{i=1}^r\ln\left(1-\alpha+\alpha\lambda_i\right)$$
$$= \sum_{i=1}^r\left[\alpha\ln\lambda_i - \ln\left(1-\alpha+\alpha\lambda_i\right)\right]$$
$$= \sum_{i=1}^r f_\alpha(\lambda_i).$$

$\square$

Now we can apply Lemma B.3 with $\Sigma = \Sigma_D^{\sqrt{b}\sigma_g}$ and $\sigma' = \Sigma_{D'}^{\sqrt{b}\sigma_g}$, then $M = \Sigma^{-\frac{1}{2}}\Sigma'\Sigma^{-\frac{1}{2}} = \left(\Sigma_D^{\sqrt{b}\sigma_g}\right)^{-\frac{1}{2}}\Sigma_{D'}^{\sqrt{b}\sigma_g}\left(\Sigma_D^{\sqrt{b}\sigma_g}\right)^{-\frac{1}{2}} = M_{D,D'}^{\sqrt{b}\sigma_g}$, and

$$D_\alpha\left(\mathcal{SGM}(\gamma(D); R, \xi)\|\mathcal{SGM}(\gamma(D'); R, \xi)\right)$$
$$= \frac{b}{2(\alpha-1)}\ln\frac{\left(\det\Sigma_D^{\sqrt{b}\sigma_g}\right)^{1-\alpha}\left(\det\Sigma_{D'}^{\sqrt{b}\sigma_g}\right)^\alpha}{\det\left((1-\alpha)\Sigma_D^{\sqrt{b}\sigma_g} + \alpha\Sigma_{D'}^{\sqrt{b}\sigma_g}\right)}$$
$$= \frac{b}{2(\alpha-1)}\sum_{i=1}^r f_\alpha(\lambda_i)$$

in which $\lambda_1 \geq \cdots \geq \lambda_r$ are the eigenvalues of $M_{D,D'}^{\sqrt{b}\sigma_g}$. Then we finish the proof. $\square$

Now we prove the monotonicity of $f_\alpha$.

**Lemma B.4** (Monotonicity of $f_\alpha$). For any $(x, \alpha)$ such that $f_\alpha$ is well-defined, $f_\alpha(x)$ is monotonically increasing with respect to $x$ when $x \geq 1$ and decreasing when $x \geq 1$.

*Proof.* By taking derivative of $f_\alpha$ with respect to $x$, we have that

$$f'(x) = \frac{\alpha}{x} - \frac{\alpha}{1 - \alpha + \alpha x} = \frac{\alpha(1 - \alpha + \alpha x) - \alpha x}{x(1 - \alpha + \alpha x)} = \frac{\alpha(1 - \alpha) - \alpha(1 - \alpha)x}{x(1 - \alpha + \alpha x)} = \frac{\alpha(\alpha - 1)(x - 1)}{x(1 - \alpha + \alpha x)}$$

so $f(x)$ is monotonically increasing for $x \geq 1$ and monotonically decreasing for $x \leq 1$. $\square$

Combining Lemma B.2 and Lemma B.4, we can get that

$$D_\alpha\left(\mathcal{SGM}(\gamma(D); R, \xi)\|\mathcal{SGM}(\gamma(D'); R, \xi)\right) = \frac{b}{2(\alpha - 1)} \sum_{i=1}^r f_\alpha(\lambda_i) \leq \frac{br}{2(\alpha - 1)} \max\{f_\alpha(\lambda_1), f_\alpha(\lambda_r)\}.$$

**Definition B.3** (Upper/Lower Ratio Sensitivity). For any constant $c \geq 0$, define the ratio sensitivity of the statistic $\gamma$ as

$$\overline{\mathrm{rsens}_c}(\gamma) = \sup_{D,D'} \lambda_{\max}\left(M_{D,D'}^c\right),$$

$$\underline{\mathrm{rsens}_c}(\gamma) = \inf_{D,D'} \lambda_{\min}\left(M_{D,D'}^c\right).$$

where the supremum is over all neighboring datasets $D, D'$.

Next we will bound the upper and lower ratio sensitivities

**Lemma B.5.**

$$1 - \frac{2\tau^2}{b\sigma_g^2} \leq \underline{\mathrm{rsens}_{\sqrt{b}\sigma_g}}(\gamma) \leq \overline{\mathrm{rsens}_{\sqrt{b}\sigma_g}}(\gamma) \leq 1 + \frac{2\tau^2}{b\sigma_g^2}.$$

*Proof.* We need to help of Weyl's inequality restated below.

**Theorem B.6** (Weyl's inequality Weyl (1912)). Let $A, B$ be Hermitian on inner product space $V$ with dimension $n$, with spectrum ordered in descending order $\lambda_1 \geq \cdots \lambda_n$. Then we have

$$\lambda_{i+j-1}(A + B) \leq \lambda_i(A) + \lambda_j(B) \leq \lambda_{i+j-n}(A + B).$$

Denote $\Delta_{D,D'} = \gamma(D')^\top \gamma(D') - \gamma(D)^\top \gamma(D)$, $E_{D,D'}^{\sqrt{b}\sigma_g} = \left(\Sigma_D^{\sqrt{b}\sigma_g}\right)^{-\frac{1}{2}} \Delta_{D,D'} \left(\Sigma_D^{\sqrt{b}\sigma_g}\right)^{-\frac{1}{2}}$, then we have

$$M_{D,D'}^{\sqrt{b}\sigma_g} = \left(\Sigma_D^{\sqrt{b}\sigma_g}\right)^{-\frac{1}{2}} \Sigma_{D'}^{\sqrt{b}\sigma_g} \left(\Sigma_D^{\sqrt{b}\sigma_g}\right)^{-\frac{1}{2}}$$

$$= \mathbb{I} + \left(\Sigma_D^{\sqrt{b}\sigma_g}\right)^{-\frac{1}{2}} \left(\Sigma_{D'}^{\sqrt{b}\sigma_g} - \Sigma_D^{\sqrt{b}\sigma_g}\right) \left(\Sigma_D^{\sqrt{b}\sigma_g}\right)^{-\frac{1}{2}}$$

$$= \mathbb{I} + \left(\Sigma_D^{\sqrt{b}\sigma_g}\right)^{-\frac{1}{2}} \left(\gamma(D')^\top \gamma(D') - \gamma(D)^\top \gamma(D)\right) \left(\Sigma_D^{\sqrt{b}\sigma_g}\right)^{-\frac{1}{2}}$$

$$= \mathbb{I} + \left(\Sigma_D^{\sqrt{b}\sigma_g}\right)^{-\frac{1}{2}} \Delta_{D,D'} \left(\Sigma_D^{\sqrt{b}\sigma_g}\right)^{-\frac{1}{2}}$$

$$= \mathbb{I} + E_{D,D'}^{\sqrt{b}\sigma_g}.$$

Since $\gamma(D)^\top \gamma(D) \succeq 0$, we have $\Sigma_D^{\sqrt{b}\sigma_g} = \gamma(D)^\top \gamma(D) + b\sigma_g^2 \mathbb{I} \succeq b\sigma_g^2 \mathbb{I}$. Therefore,

$$\left\|E_{D,D'}^{\sqrt{b}\sigma_g}\right\|_2 = \left\|\left(\Sigma_D^{\sqrt{b}\sigma_g}\right)^{-\frac{1}{2}} \Delta_{D,D'} \left(\Sigma_D^{\sqrt{b}\sigma_g}\right)^{-\frac{1}{2}}\right\|_2$$

$$\leq \left\|\left(b\sigma_g^2 \mathbb{I}\right)^{-\frac{1}{2}} \Delta_{D,D'} \left(b\sigma_g^2 \mathbb{I}\right)^{-\frac{1}{2}}\right\|_2$$

$$= \frac{\|\Delta_{D,D'}\|_2}{b\sigma_g^2}.$$

In addition,

$$\|\Delta_{D,D'}\|_2 = \left\|\gamma(D')^\top\gamma(D') - \gamma(D)^\top\gamma(D)\right\|_2$$

$$= \left\|\gamma(D')^\top\left(\gamma(D') - \gamma(D)\right) + \left(\gamma(D') - \gamma(D)\right)^\top\gamma(D)\right\|_2$$

$$\leq \|\gamma(D')\|_F \|\gamma(D') - \gamma(D)\|_F + \|\gamma(D') - \gamma(D)\|_F \|\gamma(D)\|_F$$

$$\leq 2\tau^2,$$

so we have

$$\left\|E_{D,D'}^{\sqrt{b}\sigma_g}\right\|_2 \leq \frac{\|\Delta_{D,D'}\|_2}{b\sigma_g^2} \leq \frac{2\tau^2}{b\sigma_g^2}.$$

According to Theorem B.6,

$$\left|\lambda_{\max}\left(M_{D,D'}^{\sqrt{b}\sigma_g}\right) - 1\right|_2 = \left|\lambda_{\max}\left(\mathbb{I} + E_{D,D'}^{\sqrt{b}\sigma_g}\right) - 1\right| = \left|\lambda_1\left(\mathbb{I} + E_{D,D'}^{\sqrt{b}\sigma_g}\right) - \lambda_r\left(\mathbb{I}\right)\right|$$

$$\leq \left|\lambda_{\max}\left(E_{D,D'}^{\sqrt{b}\sigma_g}\right)\right| = \left\|E_{D,D'}^{\sqrt{b}\sigma_g}\right\|_2 \leq \frac{2\tau^2}{b\sigma_g^2}.$$

Similarly,

$$\left|\lambda_{\min}\left(M_{D,D'}^{\sqrt{b}\sigma_g}\right) - 1\right|_2 \leq \frac{2\tau^2}{b\sigma_g^2}.$$

which implies

$$1 - \frac{2\tau^2}{b\sigma_g^2} \leq \lambda_{\min}\left(M_{D,D'}^{\sqrt{b}\sigma_g}\right) \leq \lambda_{\max}\left(M_{D,D'}^{\sqrt{b}\sigma_g}\right) \leq 1 + \frac{2\tau^2}{b\sigma_g^2}.$$

Since this inequality holds for all $D, D'$, we have

$$1 - \frac{2\tau^2}{b\sigma_g^2} \leq \underline{\mathrm{rsens}}_{\sqrt{b}\sigma_g}(\gamma) \leq \overline{\mathrm{rsens}}_{\sqrt{b}\sigma_g}(\gamma) \leq 1 + \frac{2\tau^2}{b\sigma_g^2}.$$

then we finish the proof. $\qquad\square$

Now we can obtain the the upper bound on $D_\alpha\left(\mathcal{SGM}(\gamma(D); R, \xi)\|\mathcal{SGM}(\gamma(D'); R, \xi)\right)$.

**Lemma B.7** (Lemma 2.7). For any pair of neighboring datasets $D, D'$,

$$D_\alpha\left(\mathcal{SGM}(\gamma(D); R, \xi)\|\mathcal{SGM}(\gamma(D'); R, \xi)\right) \leq \frac{br}{2(\alpha-1)} \max\left\{f_\alpha\left(1 + \frac{2\tau^2}{b\sigma_g^2}\right), f_\alpha\left(1 - \frac{2\tau^2}{b\sigma_g^2}\right)\right\} \leq \frac{r\alpha^2\tau^4}{(\alpha-1)b\sigma_g^4}.$$

*Proof.* With Lemma B.2 and Lemma B.4, we have

$$D_\alpha\left(\mathcal{SGM}(\gamma(D); R, \xi)\|\mathcal{SGM}(\gamma(D'); R, \xi)\right) \leq \frac{br}{2(\alpha-1)} \max\left\{f_\alpha(\lambda_1), f_\alpha(\lambda_r)\right\}.$$

According to Lemma B.5,

$$1 - \frac{2\tau^2}{b\sigma_g^2} \leq \underline{\mathrm{rsens}}_{\sqrt{b}\sigma_g}(\gamma) \leq \lambda_r \leq \lambda_1 \leq \overline{\mathrm{rsens}}_{\sqrt{b}\sigma_g}(\gamma) \leq 1 + \frac{2\tau^2}{b\sigma_g^2}$$

Applying Lemma B.4 once more, it gives that

$$D_\alpha\left(\mathcal{SGM}(\gamma(D); R, \xi)\|\mathcal{SGM}(\gamma(D'); R, \xi)\right)$$

$$\leq \frac{br}{2(\alpha-1)} \max\left\{f_\alpha(\lambda_1), f_\alpha(\lambda_r)\right\}$$

$$\leq \frac{br}{2(\alpha-1)} \max\left\{f_\alpha\left(1 + \frac{2\tau^2}{b\sigma_g^2}\right), f_\alpha\left(1 - \frac{2\tau^2}{b\sigma_g^2}\right)\right\}$$

In addition,

$$f_\alpha\left(1 + \frac{2\tau^2}{b\sigma_g^2}\right) = \alpha\ln\left(1 + \frac{2\tau^2}{b\sigma_g^2}\right) - \ln\left(1 - \alpha + \alpha\left(1 + \frac{2\tau^2}{b\sigma_g^2}\right)\right)$$

$$= \alpha \ln \left( 1 + \frac{2\tau^2}{b\sigma_g^2} \right) - \ln \left( 1 + \frac{2\alpha\tau^2}{b\sigma_g^2} \right)$$

$$\leq \alpha \cdot \frac{2\tau^2}{b\sigma_g^2} - \left( \frac{2\alpha\tau^2}{b\sigma_g^2} - \frac{1}{2} \left( \frac{2\alpha\tau^2}{b\sigma_g^2} \right)^2 \right)$$

$$= \frac{2\alpha^2\tau^4}{b^2\sigma_g^4}.$$

Similar we have

$$f_\alpha \left( 1 - \frac{2\tau^2}{b\sigma_g^2} \right) \leq \frac{2\alpha^2\tau^4}{b^2\sigma_g^4}.$$

Combining these two inequalities, we obtain

$$D_\alpha \left( \mathcal{SGM}(\gamma(D); R, \xi) \| \mathcal{SGM}(\gamma(D'); R, \xi) \right)$$

$$\leq \frac{br}{2(\alpha - 1)} \max \left\{ f_\alpha \left( 1 + \frac{2\tau^2}{b\sigma_g^2} \right), f_\alpha \left( 1 - \frac{2\tau^2}{b\sigma_g^2} \right) \right\}$$

$$\leq \frac{br}{2(\alpha - 1)} \cdot \frac{2\alpha^2\tau^4}{b^2\sigma_g^4} = \frac{r\alpha^2\tau^4}{(\alpha - 1)b\sigma_g^4},$$

then we finish the proof. $\square$

**Definition B.4** ($(\alpha, \epsilon)$-RDP Mironov (2017)). A randomized mechanism $f : \mathcal{D} \to \mathcal{R}$ is said to have $\epsilon$-Rényi differential privacy of order $\alpha$, or $(\alpha, \epsilon)$-RDP for short, if for any adjacent $D, D' \in \mathcal{D}$, it holds that

$$D_\alpha \left( f(D) \| f(D') \right) \leq \epsilon.$$

And RDP can be transformed into the standard $(\epsilon, \delta)$-DP.

**Lemma B.8** (Relationship with $(\epsilon, \delta)$-DP Mironov (2017)). If f is an $(\alpha, \epsilon)$-RDP mechanism, it also satisfies $\left( \epsilon + \frac{\ln 1/\delta}{\alpha - 1}, \delta \right)$-differential privacy for any $0 < \delta < 1$.

So we immediately have the RDP and DP result of SGM from Lemma B.7 and Lemma B.8.

**Lemma B.9** (Lemma 2.9). SGMM on the statistic $\gamma$ is $(\alpha, \frac{r\alpha^2\tau^4}{(\alpha-1)b\sigma_g^4})$-RDP, therefore $(\frac{r\alpha^2\tau^4}{(\alpha-1)b\sigma_g^4} + \frac{\ln(1/\delta)}{\alpha-1}, \delta)$-DP.

Finally we can prove Theorem 2.3.

**Theorem B.10** (Theorem 2.3). Assume $\|\gamma(D)\|_{\mathrm{F}} \leq \tau$. There exists constants $c_3$ and $c_4$ such that for for any $\epsilon_p \leq c_3$, SGMM is $(\epsilon_p, \delta_p)$-differentially privacy for any $\delta_p > 0$ if we choose

$$\sigma_g^2 \geq \frac{c_4 \sqrt{r}\tau^2 \sqrt{\ln(1/\delta_p)}}{\sqrt{b}\epsilon_p}.$$

*Proof.* According to Lemma B.9, SGMM is $(\frac{r\alpha^2\tau^4}{(\alpha-1)b\sigma_g^4} + \frac{\ln(1/\delta_p)}{\alpha-1}, \delta_p)$-DP. By taking the derivative with respect to $\alpha$, we can see the optimal choice is $\alpha = 1 + \sqrt{1 + \frac{b\sigma_g^4 \ln(1/\delta_p)}{r\tau^4}}$, then

$$\frac{r\alpha^2\tau^4}{(\alpha-1)b\sigma_g^4} + \frac{\ln(1/\delta_p)}{\alpha-1} = \frac{2r\tau^4}{b\sigma_g^4} \left( 1 + \sqrt{1 + \frac{b\sigma_g^4 \ln(1/\delta_p)}{r\tau^4}} \right)$$

in the case when $b \geq c_0 \max \left\{ \frac{r\tau^4}{\sigma_g^4 \ln(1/\delta_p)}, \frac{r\tau^4 \ln(1/\delta_p)}{\sigma_g^4} \right\}$, then $\frac{b\sigma_g^4 \ln(1/\delta_p)}{r\tau^4} \geq c_0$), we have that

$$\frac{2r\tau^4}{b\sigma_g^4} \left( 1 + \sqrt{1 + \frac{b\sigma_g^4 \ln(1/\delta_p)}{r\tau^4}} \right) \leq \frac{2r\tau^4}{b\sigma_g^4} \left( \sqrt{\frac{1}{c_0} \frac{b\sigma_g^4 \ln(1/\delta_p)}{r\tau^4}} + \sqrt{\frac{1}{c_0} \frac{b\sigma_g^4 \ln(1/\delta_p)}{r\tau^4} + \frac{b\sigma_g^4 \ln(1/\delta_p)}{r\tau^4}} \right)$$

$$= \frac{2 \left( 1 + \sqrt{c_0 + 1} \right)}{\sqrt{c_0}} \frac{\sqrt{r}\tau^2 \sqrt{\ln(1/\delta_p)}}{\sqrt{b}\sigma_g^2} := c_4 \frac{\sqrt{r}\tau^2 \sqrt{\ln(1/\delta_p)}}{\sqrt{b}\sigma_g^2}$$

so we get that SGMM on $\gamma$ is $(\epsilon_p, \delta_p)$-DP with $\epsilon_p = c_4 \frac{\sqrt{r}\tau^2 \sqrt{\ln(1/\delta_p)}}{\sqrt{b}\sigma_g^2}$. Then we finish the proof. $\square$

## B.2 Proof of Theorem 3.2

We will follow a similar analysis as in the proof of Theorem 2.3. With the same analysis in Appendix B.1, we can see that Lemma B.9 also holds for the statistic $\gamma_B^t = \sum_{c \in \mathcal{C}_t} \hat{B}_c^{t,K}$ with the sketching matrix $R_B^t$ and the random Gaussian noise $\mathbf{z}_B^t = \sum_{c \in \mathcal{C}_t} \mathbf{z}_{c,B}^t$.

**Lemma B.11.** $\mathcal{SGM}(\gamma_B^t; R_B^t, \mathbf{z}_B^t)$ is $(\alpha, \frac{r\alpha^2\tau_B^4}{(\alpha-1)b_B\sigma_{g,B}^4})$-RDP, therefore $(\frac{r\alpha^2\tau_B^4}{(\alpha-1)b_B\sigma_{g,B}^4} + \frac{\ln(1/\delta)}{\alpha-1}, \delta)$-DP.

Based on Lemma B.11, we can now prove Theorem 3.2.

**Theorem B.12** (Theorem 3.2). Denote $b_B$ the sketching dimension of SGMM on $B$, and $\sigma_{g,B}^2$ the variance of $\mathbf{z}_{c,B}^t$. There exists constants $c_7$ and $c_8$ such that given the sampling probability $q = \frac{N}{C}$ and the number of communication rounds $T$, for any $\epsilon_p \leq c_7 q\sqrt{T}$, SGMM-FFA-LoRA is $(\epsilon_p, \delta_p)$-differentially privacy for any $\delta_p > 0$ if we choose

$$\sigma_{g,B}^2 \geq \frac{c_8 q\tau_B^2 \sqrt{rT}\ln(2qT/\delta_p)}{\sqrt{b_B}\epsilon_p}.$$

*Proof.* According to Lemma B.11, $\mathcal{SGM}(\gamma_B^t; R_B^t; \mathbf{z}_B^t)$ is $\left(\frac{r\alpha^2\tau_B^4}{(\alpha-1)b_B\sigma_{g,B}^4} + \frac{\log(1/\delta_0)}{\alpha-1}, \delta_0\right)$-DP. By taking the derivative with respect to $\alpha$, we can get the optimal choice $\alpha = 1 + \sqrt{1 + \frac{b_B\sigma_{g,B}^4 \log(1/\delta_0)}{r\tau_B^4}}$, so we have

$$\frac{r\alpha^2\tau_B^4}{(\alpha-1)b_B\sigma_{g,B}^4} + \frac{\log(1/\delta_0)}{\alpha-1} = \frac{2r\tau_B^4}{b_B\sigma_{g,B}^4}\left(1 + \sqrt{1 + \frac{b_B\sigma_{g,B}^4\log(1/\delta_0)}{r\tau_B^4}}\right)$$

in the case when $b_B \geq c_0 \max\left\{\frac{r\tau_B^4}{\sigma_{g,B}^4 \log(1/\delta_0)}, \frac{r\tau_B^4 \log(1/\delta_0)}{\sigma_{g,B}^4}\right\}$, then $\frac{b_B\sigma_{g,B}^4\log(1/\delta_0)}{r\tau_B^4} \geq c_0$, we have that

$$\frac{2r\tau_B^4}{b_B\sigma_{g,B}^4}\left(1 + \sqrt{1 + \frac{b_B\sigma_{g,B}^4\log(1/\delta_0)}{r\tau_B^4}}\right)$$

$$\leq \frac{2r\tau_B^4}{b_B\sigma_{g,B}^4}\left(\sqrt{\frac{1}{c_0}\frac{b_B\sigma_{g,B}^4\log(1/\delta_0)}{r\tau_B^4}} + \sqrt{\frac{1}{c_0}\frac{b_B\sigma_{g,B}^4\log(1/\delta_0)}{r\tau_B^4} + \frac{b_B\sigma_{g,B}^4\log(1/\delta_0)}{r\tau_B^4}}\right)$$

$$= \frac{2\left(1 + \sqrt{c_0 + 1}\right)}{\sqrt{c_0}}\frac{\sqrt{r}\tau_B^2\sqrt{\log(1/\delta_0)}}{\sqrt{b_B}\sigma_{g,B}^2} := c_1 \frac{\sqrt{r}\tau_B^2\sqrt{\log(1/\delta_0)}}{\sqrt{b_B}\sigma_{g,B}^2}$$

so we get that $\mathcal{SGM}(\gamma_B^t; R_B^t, \mathbf{z}_B^t)$ is $(\epsilon_0, \delta_0)$-DP with $\epsilon_0 = c_1 \frac{\tau_B^2\sqrt{\log(1/\delta_0)}}{\sqrt{b_B}\sigma_{g,B}^2}$.

Now we list the subsampling and composition properties of $(\epsilon, \delta)$-DP.

**Lemma B.13** (Sub-sampling of $(\epsilon, \delta)$-DP). If $M$ is $(\epsilon, \delta)$-DP, then $M' = M \circ \texttt{Sample}_m$ obeys $(\epsilon', \delta')$-DP with $\epsilon' = \log(1 + p(e^\epsilon - 1))$ and $\delta' = p\delta$, in which $p = \frac{m}{n}$ is the sampling ratio.

**Lemma B.14** (Strong composition of $(\epsilon, \delta)$-DP). For all $\epsilon, \delta, \delta' \geq 0$, the class of $(\epsilon, \delta)$-differentially private mechanisms satisfies $(\epsilon', k\delta + \delta')$-differential privacy under $k$-fold adaptive composition for:

$$\epsilon' = \sqrt{2k\log(1/\delta')}\epsilon + k\epsilon(e^\epsilon - 1)$$

Since $\epsilon_0 = c_1 \frac{\sqrt{r}\tau_B^2\sqrt{\log(1/\delta_0)}}{\sqrt{b_B}\sigma_{g,B}^2} \leq \frac{c_1}{\sqrt{c_0}}$, so according to Lemma B.13, $\mathcal{SGM} \circ \texttt{Sample}_m$ satisfies $(\epsilon_1, q\delta_0)$-DP with

$$\epsilon_1 = \log(1 + q(e^{\epsilon_0} - 1)) \leq c_2 q\epsilon_0 = c_1 c_2 \frac{q\tau_B^2\sqrt{\log(1/\delta_0)}}{\sqrt{b_B}\sigma_{g,B}^2}$$

According to Lemma B.14, we can see that $T$-fold composition of $\mathcal{SGM} \circ \texttt{Sample}_m$ satisfies $(\epsilon_2, qT\delta_0 + \delta')$-DP with

$$
\begin{aligned}
\epsilon_2 &= \sqrt{2T\log(1/\delta')}\epsilon_1 + T\epsilon_1\left(e^{\epsilon_1} - 1\right) \\
&\leq \sqrt{2T\log(1/\delta')}\epsilon_1 + c_3 T\epsilon_1^2 \\
&\leq \sqrt{2T\log(1/\delta')} \cdot c_1 c_2 \frac{q\sqrt{r}\tau_B^2\sqrt{\log(1/\delta_0)}}{\sqrt{b_B}\sigma_{g,B}^2} + c_3 T \cdot \left(c_1 c_2 \frac{q\sqrt{r}\tau_B^2\sqrt{\log(1/\delta_0)}}{\sqrt{b_B}\sigma_{g,B}^2}\right)^2 \\
&= \frac{c_1 c_2\sqrt{2T\log(1/\delta')\log(1/\delta_0)}q\sqrt{r}\tau_B^2}{\sqrt{b_B}\sigma_{g,B}^2} + c_1^2 c_2^2 c_3 \frac{Tq^2 r\tau_B^4\log(1/\delta_0)}{b_B\sigma_{g,B}^4} \\
&\leq \frac{c_1 c_2 q\sqrt{2T\log(1/\delta')}}{\sqrt{c_0}} + \frac{c_1^2 c_2^2 c_3 q^2 T}{c_0}
\end{aligned}
$$

By setting $\epsilon_2 = \epsilon_p$, and choosing $\delta_0 = \frac{\delta_p}{2qT}$, $\delta' = \frac{\delta_p}{2}$,

$$
\sigma_{g,B}^2 = \frac{C'\sqrt{rT}q\tau_B^2\log(2qT/\delta_p)}{\sqrt{b_B}\epsilon_p},
$$

then we can obtain $(\epsilon_p, \delta_p)$-DP. $\qquad\square$

### B.3 PROOF OF THEOREM 3.3

**Theorem B.15** (Theorem 3.3). Denote $b_A$ and $b_B$ the sketching dimensions of SGMV on $A$ and $B$, and $\sigma_{g,A}^2$ and $\sigma_{g,B}^2$ the variance of $\mathbf{z}_{c,A}^t$ and $\mathbf{z}_{c,B}^t$. There exists constants $c_7$ and $c_8$ such that given the sampling probability $q = \frac{N}{C}$ and the number of communication rounds $T$, for any $\epsilon_p \leq c_7 q\sqrt{T}$, SGMM-FlexLoRA is $(2\epsilon_p, 2\delta_p)$-differentially privacy for any $\delta_p > 0$ if we choose

$$
\sigma_{g,A}^2 \geq \frac{c_8 q\tau_A^2\sqrt{rT}\ln(2qT/\delta_p)}{\sqrt{b_A}\epsilon_p}, \quad \sigma_{g,B}^2 \geq \frac{c_8 q\tau_B^2\sqrt{rT}\ln(2qT/\delta_p)}{\sqrt{b_B}\epsilon_p}.
$$

*Proof.* Following the same analysis as in the proof of Theorem B.12, we can see that for any $\epsilon_p \leq c_7 q\sqrt{T}$, SGMM on $\gamma_A^t = \sum_{c \in \mathcal{C}_t} \hat{A}_c^{t,K}$ with the sketching matrix $R_A^t$ and the random Gaussian noise $\mathbf{z}_A^t = \sum_{c \in \mathcal{C}_t} \mathbf{z}_{c,A}^t$ is $(\epsilon_p, \delta_p)$-differentially private for any $\delta_p > 0$ if we choose

$$
\sigma_{g,A}^2 \geq \frac{c_8 q\tau_A^2\sqrt{rT}\ln\left(2qT/\delta_p\right)}{\sqrt{b_A}\epsilon_p}.
$$

And SGMM on $\gamma_B^t = \sum_{c \in \mathcal{C}_t} \hat{B}_c^{t,K}$ with the sketching matrix $R_B^t$ and the random Gaussian noise $\mathbf{z}_B^t = \sum_{c \in \mathcal{C}_t} \mathbf{z}_{c,B}^t$ is $(\epsilon_p, \delta_p)$-differentially private for any $\delta_p > 0$ if we choose

$$
\sigma_{g,B}^2 \geq \frac{c_8 q\tau_B^2\sqrt{rT}\ln\left(2qT/\delta_p\right)}{\sqrt{b_B}\epsilon_p}.
$$

Then Theorem B.15 can be immediately produced by invoking the following composition theorem with SGMM on adapter matrices $A$ and $B$.

**Theorem B.16** (Dwork et al. (2014)). Let $\mathcal{M}_i : \mathbb{N}^{|\mathcal{X}|} \to \mathcal{R}_i$ be an $(\varepsilon_i, \delta_i)$-differentially private algorithm for $i \in [k]$. Define $\mathcal{M}_{[k]} : \mathbb{N}^{|\mathcal{X}|} \to \prod_{i=1}^{k} \mathcal{R}_i$ by

$$
\mathcal{M}_{[k]}(x) = \left(\mathcal{M}_1(x), \ldots, \mathcal{M}_k(x)\right).
$$

Then $\mathcal{M}_{[k]}$ is $\left(\sum_{i=1}^{k} \varepsilon_i, \sum_{i=1}^{k} \delta_i\right)$-differentially private.

$\qquad\square$

