# OpenReview forum: "Sketched Gaussian Mechanism on Matrix for Private Federated LoRA"
_ICLR.cc/2026/Conference — Submitted to ICLR 2026_

### Official Review · Reviewer_yPFM · 2025-10-25

**Soundness:** 2
**Presentation:** 2
**Contribution:** 2
**Rating:** 4
**Confidence:** 2

**Summary:**

This paper proposes a mechanism named Sketched Gaussian Mechanism on Matrix (SGMM) for private federated LoRA. By combining random sketching with the Gaussian mechanism at the matrix level , this method aims to simultaneously address two challenges in federated LoRA: the high communication overhead from large adapter factors and the significant noise amplification caused by standard DP mechanisms. The authors provide a theoretical RDP analysis of SGMM and integrate it into the FFA-LORA and FlexLoRA algorithms.

**Strengths:**

This paper is the first to integrate matrix sketching methods with the federated LoRA, supported by a rigorous privacy guarantee using the Rényi Differential Privacy framework.

**Weaknesses:**

The paper's novelty is constrained, as the core matrix sketching-plus-noise mechanism exists in prior works such as [1], and recent research offers independent RDP analyses of this technique [2]. Furthermore, the empirical support presented is limited, focusing narrowly on a single dataset with fixed privacy and sketching parameters. Broader experiments covering diverse datasets, sensitivity analyses across key parameters, and evaluations of efficiency metrics are needed to convincingly establish practical benefits.

[1] Yuchang Sun, Jiawei Shao, Songze Li, Yuyi Mao, and Jun Zhang, "Stochastic Coded Federated Learning with Convergence and Privacy Guarantees." 2022 IEEE International Symposium on Information Theory (ISIT), pages 2028-2033, 2022.
[2] Omri Lev, Vishwak Srinivasan, Katrina Ligett, Ayush Sekhari, and Ashia C Wilson, "The Gaussian Mixing Mechanism: Renyi Differential Privacy via Gaussian Sketches." Accepted to the 38th Advances in Neural Information Processing Systems (NeurIPS 2025), 2025.

**Questions:**

1. How does the theoretical RDP bound derived in Theorem 2.3 compare in tightness to privacy analyses of similar client-level sketching-plus-noise mechanisms, such as the MI-DP analysis in work [1] ?
2. The derived privacy bound suggests noise variance scales approximately as $\sqrt{r}/\sqrt{b}$. Achieving a small privacy loss $\epsilon_p$ thus seems to require $b$ to be comparable to $r$, implying substantial noise if the rank $r$ is large while a strong privacy guarantee (small $\epsilon_p$) is desired. Does this indicate a potential practicality issue for SGMM when applied to high-rank LoRA adaptations under strict privacy constraints?
3. How does model performance, including accuracy and convergence, trade off across varying privacy budgets $\epsilon_p$, sketch dimensions $b$, and LoRA ranks $r$?

[1] Yuchang Sun, Jiawei Shao, Songze Li, Yuyi Mao, and Jun Zhang, "Stochastic Coded Federated Learning with Convergence and Privacy Guarantees." 2022 IEEE International Symposium on Information Theory (ISIT), pages 2028-2033, 2022.

---

> ### Author Response · Authors · 2025-12-01
> **Response to Reviewer yPFM**
>
> **Q1:** "The paper's novelty is constrained, as the core matrix sketching-plus-noise mechanism exists in prior works such as [1], and recent research offers independent RDP analyses of this technique [2]."
>
> "How does the theoretical RDP bound derived in Theorem 2.3 compare in tightness to privacy analyses of similar client-level sketching-plus-noise mechanisms, such as the MI-DP analysis in work [1]?"
>
> [1] Yuchang Sun, Jiawei Shao, Songze Li, Yuyi Mao, and Jun Zhang, "Stochastic Coded Federated Learning with Convergence and Privacy Guarantees." 2022 IEEE International Symposium on Information Theory (ISIT), pages 2028-2033, 2022.
>
> [2] Omri Lev, Vishwak Srinivasan, Katrina Ligett, Ayush Sekhari, and Ashia C Wilson, "The Gaussian Mixing Mechanism: Renyi Differential Privacy via Gaussian Sketches." Accepted to the 38th Advances in Neural Information Processing Systems (NeurIPS 2025), 2025.
>
> We fully acknowledge that both [1] and [2] also study mechanisms that combine sketching with additive (Gaussian) noise, and that [2] in particular employs Rényi DP tools for its analysis. However, our work differs from these papers in both the privacy notion and the learning scenario, which leads to a distinct technical focus and result.
>
> (i) Comparison to [1]: different privacy notion and behavior in $b$.
>
> Our paper focuses on client-level differential privacy in federated learning, where the requirement is that the server cannot reliably infer the participation of any single client by observing the privatized local updates. In contrast, [1] also considers a federated learning setting (for linear regression), but adopts mutual information DP (MI-DP) as defined in their Definition 1, which is quite different from the client-level DP notion we study. Accordingly, their privacy guarantee in Theorem 2 is formulated in terms of MI-DP, not client-level $(\epsilon,\delta)$-DP.
>
> Furthermore, the MI-DP bound in [1] does not exhibit a monotone improvement with respect to the sketching dimension in the way our bound does. In our Theorem 2.3, the derived client-level DP bound is explicitly and monotonically decreasing in the sketching dimension $b$, making clear how increasing $b$ reduces the required noise variance. This explicit and monotone dependence on the sketching dimension is not captured by the MI-DP analysis in [1], so the two results are not directly comparable in terms of “tightness” under the same privacy definition. Instead, our contribution is to provide a new, explicit, client-level DP characterization tailored to federated learning with matrix-valued updates.
>
> (ii) Comparison to [2]: different adjacency notion and explicit dependence on $b$.
>
> The analysis in [2] is indeed closer in spirit to ours, in that it uses Rényi DP techniques for a Gaussian sketching–based mechanism. However, [2] focuses on centralized linear and logistic regression, and consequently considers a row-level adjacency model: as stated on their page 4, neighboring datasets differ in a single row of the data matrix $X$ (or equivalently, the matrix passed through the sketching step differs in one row). Under this setting, the sketching matrix multiplies a data matrix with a very structured notion of neighboring datasets, and the resulting RDP bounds are not expressed with an explicit dependence on the sketching dimension $b$.
>
> By contrast, we consider client-level DP in federated learning, where the matrix $\gamma(D)$ passed through sketching is itself the (matrix-valued) output of a local learning algorithm run on the client’s dataset $D$. For neighboring datasets $D$ and $D'$, the corresponding matrices $\gamma(D)$ and $\gamma(D')$ can differ across the entire matrix, not just in a single row. Our main theoretical result (Theorem 2.3) shows that, even under this much more general adjacency model, one can still obtain a client-level DP bound with an explicit and monotone dependence on the sketching dimension $b$.

---

> ### Author Response · Authors · 2025-12-01
> **Response to Reviewer yPFM**
>
> **Q2:** "Furthermore, the empirical support presented is limited, focusing narrowly on a single dataset with fixed privacy and sketching parameters. Broader experiments covering diverse datasets, sensitivity analyses across key parameters, and evaluations of efficiency metrics are needed to convincingly establish practical benefits."
>
> "How does model performance, including accuracy and convergence, trade off across varying privacy budgets $\epsilon_{p}$, sketching dimensions $b$, and LoRA ranks $r$?"
>
> We thank the reviewer for these suggestions. Our work is primarily theoretical: the main goal is to formalize and explain the phenomenon that sketching itself can provide privacy, and that sketching-based mechanisms can require less Gaussian noise than their non-sketched counterparts to achieve the same privacy level, now extended from vector mechanisms to matrix-valued algorithms. LoRA is used in our paper mainly as a concrete and practically relevant application of this matrix-level theory. The experiments are therefore intended as illustrative evidence that sketching-based algorithms can still deliver competitive, and sometimes even better, accuracies than the corresponding non-sketched private baselines, rather than as a comprehensive empirical study over all parameter ranges.
>
> We provide an additional experiment example. We examine the setting where the privacy level is $\epsilon_{p}=0.75$, while keeping all other parameters, the model, and the dataset the same as in the experiments in the paper. In this case, the non-sketching version of private FFA-LoRA needs a noise multiplier of 3, SGMM-FFA-LoRA needs a noise multiplier of 2.0086, and SGMV-FFA-LoRA needs a noise multiplier of 1.0045, and we obtain the following accuracies:
>
> | Algorithm                          | private FFA-LoRA (no sketching) | SGMM-FFA-LoRA | SGMV-FFA-LoRA |
> |------------------------------------|----------------------------------|---------------|---------------|
> | Accuracy                           | 0.4550                           | 0.4958        | 0.5256        |
>
> We can see that under this setting, the two sketching versions, SGMM-FFA-LoRA and SGMV-FFA-LoRA, both outperform the non-sketching version. We will add broader experiments covering diverse datasets, systematic sensitivity analyses across $\epsilon_{p}$, $b$, and $r$, and evaluations of efficiency metrics in the revised version.
>
> ---
>
> **Q3:** "The derived privacy bound suggests noise variance scales approximately as $\sqrt{r}/\sqrt{b}$. Achieving a small privacy loss $\epsilon_{p}$ thus seems to require $b$ to be comparable to $r$, implying substantial noise if the rank $r$ is large while a strong privacy guarantee (small $\epsilon_{p}$) is desired. Does this indicate a potential practicality issue for SGMM when applied to high-rank LoRA adaptations under strict privacy constraints?"
>
> From a practical perspective, LoRA is specifically designed to parameterize task-specific updates via a low-rank factorization $AB$, which dramatically reduces the number of trainable parameters and the memory/optimizer overhead. Consequently, in realistic applications the rank $r$ is chosen to be much smaller than the layer dimensions $d$ and $k$; typical choices are $r\leq32$ or $64$ even for very large models. If $r$ were comparable to the original dimension, the parameter-efficiency and computational advantages of LoRA would largely disappear, and one might as well fine-tune the full matrix. In this realistic low-rank regime, there is ample room to choose a sketching dimension $b$ that is still moderately large compared to $r$, so that the ratio $\sqrt{r}/\sqrt{b}$ remains small.
>
> It is conceptually unavoidable that achieving a very strong privacy guarantee (small $\epsilon_{p}$) requires injecting substantial noise, regardless of the mechanism used. Our goal in this work is therefore comparative: we compare the amount of noise required by SGMM to that required by the classical GM, and to show that SGMM can strictly reduce this noise for the same $(\epsilon_{p},\delta_{p})$. Under the empirically reality that $r\ll d,k$ and that $b$ is chosen with a reasonable compression rate, Remark 2.1 shows precisely that SGMM needs less noise than GM to reach the same privacy level.
>
> Moreover, as model sizes grow, practitioners typically keep $r$ small (e.g., 32 or 64) while the layer dimensions $d,k$ increase, which allows $b$ to scale up as well. In this large-model regime, the condition $b\gg r$ becomes even easier to satisfy, and the practicality and advantage of SGMM over GM become more pronounced rather than diminished.

---

### Official Review · Reviewer_9JDd · 2025-10-26

**Soundness:** 3
**Presentation:** 2
**Contribution:** 3
**Rating:** 6
**Confidence:** 3

**Summary:**

The paper proposes the Sketched Gaussian Mechanism on Matrix (SGMM), an adapted approach of the earlier proposed SGMV method, to improve communication efficiency and privacy in federated Low-Rank Adaptation (LoRA) for large language models. It is shown that the proposed algorithm achieves the same privacy protection strength with noise magnitude of $1/\sqrt{b}$ ($b$ is the dimension of sketch), which can be better than the vanilla Gaussian mechanism. The authors combine SGMM with the existing federated learning LoRA algorithms and demonstrate some empirical results as well.

**Strengths:**

1. The paper proposed an improved version of sketching algorithms for matrices, SGMM. The algorithm is insightful and well-motivated.
2. The paper presents a detailed analysis of the privacy proofs of the proposed algorithm.
2. The paper provides some necessary empirical analysis to show the effectiveness of the proposed algorithm.

**Weaknesses:**

1. The proposed algorithm relies on SVD of a matrix, which will introduce additional computation and could be hard to scale to large model.
2. Although the authors show the benefit of the proposed algorithm has a certain level of theoretical benefit, their empirical results show that such a benefit might be very limited when such an algorithm is applied to model training.
3. The writing of the paper can be further improved. The readers may find it easy to get lost in what the main contributions are in the paper.
4. There is no utility analysis of the proposed algorithm. While the privacy proof is presented, it is not clear how the magnitude of noise can theoretically affect the (reconstructed) matrices, e.g., the $\tilde{B}^{t, k}_c$.

**Questions:**

1. Is it possible to provide some theoretical analysis about how the noise and sketch dimension can affect the utility?
2. What will be the computation overhead (considering SVD) when the SGMM is applied to model training?
3. What can be the potential reason that SGMM-FFA-LoRA and SGMM-FlexLoRA do not show significant performance gain? Is it related to the sketch/matrix dimension?

---

> ### Author Response · Authors · 2025-12-01
> **Response to Reviewer 9JDd**
>
> **Q1:** "The proposed algorithm relies on SVD of a matrix, which will introduce additional computation and could be hard to scale to large model."
>
> "What will be the computation overhead (considering SVD) when the SGMM is applied to model training?"
>
> Thank you for raising this concern. Our SGMM-FlexLoRA algorithm is built on top of the original FlexLoRA method, and the SVD step we use is exactly the same as in FlexLoRA. As stated in the FlexLoRA paper, “Compared to FedAvg and homogeneous rank-based FL methods, FlexLoRA incorporates a lightweight SVD procedure, but the overhead from SVD is negligible compared to the local LLM training procedure. Moreover, the SVD is performed only once per round and is independent of client numbers.” Thus, even before adding SGMM, the computational cost of the SVD step is relatively small compared with client-side training.
>
> Concretely, in the original FlexLoRA, the aggregated LoRA weight matrix at round $t+1$ is $\tilde{W}^{t+1}\in\mathbb{R}^{d\times k}$, and a truncated rank-$r$ SVD of $\tilde{W}^{t+1}$ has per-round complexity $\mathcal{O}(dkr)$ on the server. With the combination of sketching in SGMM, we effectively replace $\tilde{W}^{t+1}$ with its sketched version $W^{t+1}\in\mathbb{R}^{b_{B}\times b_{A}}$, where $b_{A}\ll k$ and $b_{B}\ll d$. The SVD is now performed on $W^{t+1}$, so the computation cost becomes $\mathcal{O}(b_{A}b_{B}r)$, which is strictly smaller than $\mathcal{O}(dkr)$ when $b_{A},b_{B}$ are chosen to be much smaller than $k,d$.
>
> Even after accounting for the additional matrix multiplications introduced by sketching on the clients (i.e., forming matrix products $A^{t}R_{A}^{t}$ and $B^{t}R_{B}^{t}{}^{\top}$), the overall per-round overhead due to SGMM and SVD is on the order of $\mathcal{O}(b_{A}b_{B}r)+\mathcal{O}(db_{B}r+kb_{A}r)$, where the first term is the SVD on the sketched matrix and the second term corresponds to the sketching multiplications. This cost is still substantially smaller than the original $\mathcal{O}(dkr)$ SVD when $b_{A},b_{B}\ll k,d$, and, in practice, remains negligible compared to the cost of local LLM training on each client. Moreover, all SVD computations are carried out on the server, which typically has much more abundant computational resources than the clients in federated learning.
>
> ---
>
> **Q2:** "Although the authors show the benefit of the proposed algorithm has a certain level of theoretical benefit, their empirical results show that such a benefit might be very limited when such an algorithm is applied to model training."
>
> "What can be the potential reason that SGMM-FFA-LoRA and SGMM-FlexLoRA do not show significant performance gain? Is it related to the sketch/matrix dimension?"
>
> We thank the reviewer for this comment. When comparing the SGMM and SGMV versions of the algorithm with their non-sketched counterparts, whether the sketching versions can outperform is highly dependent on the choice of hyperparameters. According to Remark 2.1, when $b\geq\Omega\left(\frac{r\epsilon_{p}^{2}}{\ln\left(1/\delta_{p}\right)}\right)$, the noise needed by SGMM is smaller than that required by the classical GM. However, sketching itself can introduce some degradation to the performance of the original algorithm, which may offset the gain from noise reduction. From the comparison in Remark 2.1 we can also see that the reduction in noise variance is more apparent when the sketching dimension $b$ (or, with a fixed compression rate, the original dimension $d$) is large, or when the privacy level $\epsilon_{p}$ is small.
>
> For example, we examine the setting where the privacy level is $\epsilon_{p}=0.75$, while keeping all other parameters, the model, and the dataset the same as in the experiments in the paper. In this case, the non-sketching version of private FFA-LoRA needs a noise multiplier of 3, SGMM-FFA-LoRA needs a noise multiplier of 2.0086, and SGMV-FFA-LoRA needs a noise multiplier of 1.0045, and we obtain the following accuracies:
>
> | Algorithm                          | private FFA-LoRA (no sketching) | SGMM-FFA-LoRA | SGMV-FFA-LoRA |
> |------------------------------------|----------------------------------|---------------|---------------|
> | Accuracy                           | 0.4550                           | 0.4958        | 0.5256        |
>
> We can see that under this setting, the two sketching versions, SGMM-FFA-LoRA and SGMV-FFA-LoRA, both outperform the non-sketching version. We will add more experiments on different tasks and models in the future revision.

---

> ### Author Response · Authors · 2025-12-01
> **Response to Reviewer 9JDd**
>
> **Q3:** "The writing of the paper can be further improved. The readers may find it easy to get lost in what the main contributions are in the paper."
>
> We appreciate this feedback on the presentation of the paper. In the revision, we will improve the exposition and organization, especially in the introduction and conclusion, to make the main contributions easier to follow.
>
> Specifically, our main contribution is to show that the sketched Gaussian mechanism (SGM) for vectors can be generalized to matrices as SGMM, while preserving the key property that, when the sketching dimension is relatively large, SGMM requires strictly less added Gaussian noise than the classical Gaussian mechanism (GM) to achieve the same privacy level. This demonstrates that the inherent privacy provided by sketching continues to hold in the matrix setting. In addition, we show how this mechanism can be applied to LoRA, a widely used low-rank matrix adaptation method. In our experiments, we confirm that, in the considered setups, the sketched private variants indeed need strictly less added noise than GM-based baselines to reach the same privacy level, while maintaining competitive empirical performance.
>
> ---
>
> **Q4:** "There is no utility analysis of the proposed algorithm. While the privacy proof is presented, it is not clear how the magnitude of noise can theoretically affect the (reconstructed) matrices, e.g., the $\tilde{B}_{c}^{t,k}$."
>
> "Is it possible to provide some theoretical analysis about how the noise and sketching dimension can affect the utility?"
>
> We thank the reviewer for raising this important point. We agree that a formal utility analysis, quantifying how the injected noise and the sketching dimensions affect the reconstructed matrices (e.g., $\tilde{B}_{c}^{t,k}$), would further strengthen the work. However, obtaining such a result would require convergence/optimization guarantees for the underlying algorithms. Our private algorithms SGMM/SGMV-FFA-LoRA and SGMM-FlexLoRA are direct private counterparts of FFA-LoRA and FlexLoRA, and to the best of our knowledge, the original works on FFA-LoRA and FlexLoRA do not provide optimization or convergence analyses. As a result, deriving rigorous utility bounds for our private variants would essentially require first developing a full convergence theory for these base methods, which is a technically substantial and orthogonal undertaking.
>
> In this paper, we therefore chose to focus on the privacy properties of the proposed sketched mechanisms and on demonstrating that they can be integrated into these practical LoRA-based FL algorithms while maintaining good empirical performance. We view establishing a theoretical utility/convergence analysis that explicitly characterizes the effect of the sketching dimension and noise magnitude as an important and interesting direction for future work.

---

### Official Review · Reviewer_8FnX · 2025-11-03

**Soundness:** 3
**Presentation:** 3
**Contribution:** 2
**Rating:** 2
**Confidence:** 4

**Summary:**

This paper proposes the Sketched Gaussian Mechanism on Matrix (SGMM), which couples random sketching with the Gaussian mechanism at the matrix level. It also provides a unified privacy analysis of the proposed sketching mechanism, which shows that, for a fixed privacy level, the required noise variance scales inversely proportional to the sketch dimension. Finally, they apply the idea to the setting of federated low-rank adaptation, motivated by reducing the communication overhead and achieving client-level privacy.

**Strengths:**

1. The paper studies the trilemma between communication overhead, privacy and utility in federated low-rank adaptation, which is an interesting and important problem.
2. The work proposes a sketching approach natively designed for matrix statistics, which is an important problem and improves the SoTA of sketching approaches.
3. The authors also provide the theoretical privacy analysis of the proposed approach, showing its difference with that of the existing vectorization-based sketching approaches.

**Weaknesses:**

1. The theoretical results in this work suggest the following main messages:

-  For a fixed privacy level, the order of Gaussian noise magnitude is: GM > SGMM > SGMV (with $h=br$)
- SGMM has less computational complexity than SGMV.
- Using SGMM and SGMV can reduce the communication overhead in federated low-rank adaptation, but potentially at the cost of model utility.

However, the experimental results are very limited and do not fully support the above claims/findings. Further questions are asked about this below.

2. An important limitation of the proposed method when combined with FFA-LoRA is that all the participating clients in federated low-rank adaptation need to use the same sketch matrices $R_B^t$ (as well as $R_A^t$ for Flex-LoRA) in each round $t$. This is a strong limitation, as clients in FL cannot communicate to synchronize their matrices. Also, even the server cannot be aware of the matrices, as from the client-level privacy considered in the paper, it seems that the server is not trusted. Even in algorithm 1 and 2, it is not clear where the sketch matrices in each round $t$ come from.

3. While the work has some contributions, its main achievements seem unclear.

Overall, the work needs to get improved, especially the experimental results.

**Questions:**

Following the weaknesses mentioned above, I have the following questions:

1. It has been shown that for a fixed privacy level, SGMM adds more Gaussian noise, and has less computational complexity than SGMV. However, in Fig. 1 (a), SGMM seems to get a better utility than SGMV (on average). Is this result inconsistent with the findings mentioned above?

2. Similarly, in Fig. 1 (b), GM clearly performs better than SGMM. SGMM adds less noise than GM, and has some utility loss due to incorporating the random sketching. Considering Fig. 1 (a) and (b), it seems that we cannot make a clear conclusion when comparing the utility of GM with that of SGMV and SGMM. In other words, they get comparable utility. So, at a fixed privacy level, what is gained from using sketching (either SGMV or SGMM)? Only reducing the communication overhead?

3. In algorithms 1 and 2, the sketch matrices of participating clients in each round $t$, should be synchronized. Have the authors considered this important point?

minor comments:

typo in line 315: analogously

typo in line 348: $R_A^t$ should be $R_A^{t^T}$?

---

> ### Author Response · Authors · 2025-12-01
> **Response to Reviewer 8FnX**
>
> **Q1:** "The theoretical results in this work suggest the following main messages:
> - For a fixed privacy level, the order of Gaussian noise magnitude is: GM$>$SGMM$>$SGMV (with $h=br$).
> - SGMM has less computational complexity than SGMV.
> - Using SGMM and SGMV can reduce the communication overhead in federated low-rank adaptation, but potentially at the cost of model utility.
> However, the experimental results are very limited and do not fully support the above claims/findings."
>
> "It has been shown that for a fixed privacy level, SGMM adds more Gaussian noise, and has less computational complexity than SGMV. However, in Fig. 1 (a), SGMM seems to get a better utility than SGMV (on average). Is this result inconsistent with the findings mentioned above?"
>
> "Similarly, in Fig. 1 (b), GM clearly performs better than SGMM. SGMM adds less noise than GM, and has some utility loss due to incorporating the random sketching. Considering Fig. 1 (a) and (b), it seems that we cannot make a clear conclusion when comparing the utility of GM with that of SGMV and SGMM. In other words, they get comparable utility. So, at a fixed privacy level, what is gained from using sketching (either SGMV or SGMM)? Only reducing the communication overhead? "
>
> We thank the reviewer for this comment. When comparing the SGMM and SGMV versions of the algorithm with their non-sketched counterparts, whether the sketching variants can outperform is highly dependent on the choice of hyperparameters. According to Remark 2.1, when $b\geq\Omega\left(\frac{r\epsilon_{p}^{2}}{\ln\left(1/\delta_{p}\right)}\right)$, the noise needed by SGMM is smaller than that required by the classical GM. However, because sketching can harm the underlying model’s accuracy, its negative impact may partially cancel out the improvement gained by reducing the noise level. In addition, we can also see from the comparison in Remark 2.1 that the reduction in noise variance is more apparent when the sketching dimension $b$ (or, with a fixed compression rate, the original dimension $d$) is large, or when the privacy level $\epsilon_{p}$ is small.
>
> For example, we examine the setting where the privacy level is $\epsilon_{p}=0.75$, while keeping all other parameters, the model, and the dataset the same as in the experiments in the paper. In this case, the non-sketched version of private FFA-LoRA requires a noise multiplier of 3, SGMM-FFA-LoRA requires a noise multiplier of 2.0086, and SGMV-FFA-LoRA requires a noise multiplier of 1.0045, and we obtain the following accuracies:
>
> | Algorithm                          | private FFA-LoRA (no sketching) | SGMM-FFA-LoRA | SGMV-FFA-LoRA |
> |------------------------------------|----------------------------------|---------------|---------------|
> | Accuracy                           | 0.4550                           | 0.4958        | 0.5256        |
>
> We can see that under this setting, the two sketching versions, SGMM-FFA-LoRA and SGMV-FFA-LoRA, both outperform the non-sketched version. We will add more experiments on different tasks and models in the revised version.
>
> In addition, although Theorem 3.1 and Theorem 3.2 show that the noise variance required by SGMV-FFA-LoRA is smaller than that required by SGMM-FFA-LoRA, the sketching Gaussian matrices in SGMV-FFA-LoRA and SGMM-FFA-LoRA have different dimensions and variances. As a result, the effect of sketching on performance can also differ between the two algorithms, and it may happen that SGMM-FFA-LoRA attains comparable or even better empirical performance than SGMV-FFA-LoRA. Therefore, the fact that SGMM-FFA-LoRA performs better in our Figure 1 does not contradict our theoretical results.

---

> ### Author Response · Authors · 2025-12-01
> **Response to Reviewer 8FnX**
>
> **Q2:** "An important limitation of the proposed method when combined with FFA-LoRA is that all the participating clients in federated low-rank adaptation need to use the same sketch matrices $R_{B}^{t}$ (as well as $R_{A}^{t}$ for Flex-LoRA) in each round $t$. This is a strong limitation, as clients in FL cannot communicate to synchronize their matrices. Also, even the server cannot be aware of the matrices, as from the client-level privacy considered in the paper, it seems that the server is not trusted. Even in algorithm 1 and 2, it is not clear where the sketch matrices in each round $t$ come from."
>
> "In algorithms 1 and 2, the sketch matrices of participating clients in each round $t$, should be synchronized. Have the authors considered this important point?"
>
> We thank the reviewer for highlighting the need to clarify how the sketch matrices are synchronized across clients. In our intended deployment, the sketching matrices $R_{A}^{t}$ and $R_{B}^{t}$ are generated from a shared secret random seed that is known to all participating clients but not to the server. Concretely, we assume an initialization phase (e.g., the same type of key-setup phase that is already standard in secure aggregation–based FL systems) during which clients obtain a common secret seed $\texttt{seed}\_{A}$ and $\texttt{seed}\_{B}$. In each round $t$, every client locally computes
> \begin{align*}
>     R_{A}^{t}=\text{PRG}\left(\texttt{seed}\_{A}, t\right), R_{B}^{t}=\text{PRG}\left(\texttt{seed}\_{B}, t\right)
> \end{align*}
> where $\text{PRG}$ is a pseudorandom generator that maps the seed and round index to an i.i.d. Gaussian matrix. This guarantees that (i) all clients use exactly the same sketch matrices in round $t$, without any client–client communication, and (ii) the server, which does not know $\texttt{seed}$, cannot reconstruct these matrices. We will make this assumption and the corresponding construction explicit in Algorithms 1 and 2.
>
> ---
>
> **Q3:** "While the work has some contributions, its main achievements seem unclear."
>
> We thank the reviewer for this comment and agree that the presentation of our main achievements can be improved. In the revision, we will refine the exposition and structure, particularly in the introduction and conclusion, to highlight our key contributions more clearly.
>
> In summary, our primary achievement is to generalize the sketched Gaussian mechanism (SGM) from vectors to matrices, resulting in SGMM, and to show that it preserves the crucial property that, when the sketching dimension is relatively large, SGMM requires strictly less added Gaussian noise than the classical Gaussian mechanism (GM) to achieve the same privacy level. This establishes that the inherent privacy benefits of sketching extend from the vector case to the matrix setting. Moreover, we demonstrate how this mechanism can be integrated into LoRA, a widely used low-rank matrix adaptation method. Empirically, in our experimental setups, the sketched private variants consistently require strictly less added noise than GM-based baselines to attain the same privacy budget, while still achieving competitive performance.
>
> ---
>
> **Minor1:** "typo in line 315: analogously"
>
> Thank you for pointing out this typo. We will correct it in the revision and carefully proofread the manuscript to fix any similar typographical errors.
>
> ---
>
> **Minor2:** "typo in line 348: $R_{A}^{t}$ should be $R_{A}^{t}{}^{\top}$?"
>
> Thank you for pointing this out. According to our setting, $A^{t}\in\mathbb{R}^{r\times k}$ and $B^t\in\mathbb{R}^{d\times r}$. We choose to write the sketching matrices in the form that their dimensions are always “original dimension
> $\times$ sketching dimension,”, i.e., $R_{A}^{t}\in\mathbb{R}^{k\times b_{A}}$ and $R_{B}^{t}\in\mathbb{R}^{d\times b_{B}}$. Consequently, to make the matrix multiplications dimensionally consistent, the correct forms should be $A^{t}R_{A}^{t}$ and $R_{B}^{t}{}^{\top}B^{t}$. We will clarify this point in the revised version and correct the notation to avoid confusion.

---

### Official Review · Reviewer_YSJc · 2025-11-03

**Soundness:** 3
**Presentation:** 3
**Contribution:** 2
**Rating:** 4
**Confidence:** 2

**Summary:**

The paper proposes a Sketched Gaussian Matrix Mechanism (SGMM) for federated LoRA. The proposed SGMM is specifically designed to do DP for matrix data instead of the conventional vector data. Theoretical analysis shows the condition under which the proposed SGMM is better than the classical GM, which basically translates to a sufficiently small rank $r$. Empirical results on CIFAR 100 and ViT demonstrate the effectiveness of the proposed method.

**Strengths:**

This paper is based on a clean and important motivation: DP in federated LoRA. Given that classical DP uses Gaussian mechanisms for vectors, the proposed sketched Gaussian mechanism for matrix is a reasonable step. The theoretical analysis justifies the advantages of the proposed method under low rank adaptation.

**Weaknesses:**

* Although theoretical analysis seems sound, it is an incremental step from the existing knowledge (e.g., Theorem 1 and standard sketched Gaussian Mechanism).
* Given the limited theoretical contribution, a strong empirical contribution is expected. However, this paper only evaluate the proposed method on CIFAR100 under a single privacy setting. The experiments only use ViT, while the paper motivates itself using LLMs. Moreover, the empirical results do not show advantage of the proposed method comparing to standard Gaussian mechanisms, in contrast to what the theory might suggest.
* Minor issue in Line 110: citation Dwork et al. (2006) should use another format.

**Questions:**

* Remark 2.1 states that "Comparing to classical GM, SGMM attains the same privacy level with smaller noise whenever the sketch dimension satisfies $b\geq \Omega (\frac{r\epsilon_p^2}{\ln(1/\delta_p)})$." However, if we have $r=1$ and the matrix reduce to a vector, how the classical GM is better than the SGMM? I may misunderstand this point, and an explanation is appreciated.

---

> ### Author Response · Authors · 2025-12-01
> **Response to Reviewer YSJc**
>
> **Q1:** "Although theoretical analysis seems sound, it is an incremental step from the existing knowledge (e.g., Theorem 1 and standard sketched Gaussian Mechanism)."
>
> Our main theoretical contribution is to extend SGM from the vector setting to the matrix setting, yielding SGMM, while rigorously showing that it preserves the key property that, when the sketching dimension is relatively large, SGMM requires strictly less added Gaussian noise than the classical Gaussian mechanism (GM) to achieve the same privacy level. Establishing this generalization is not a direct corollary of prior work: it requires a more delicate analysis of the matrix spectrum (e.g., via Weyl’s inequality and related tools), which is specific to the matrix-valued setting. This demonstrates that the inherent privacy benefit of sketching continues to hold beyond vectors and remains valid for matrices.
>
> In addition, we show how this mechanism can be integrated into LoRA, a widely used low-rank matrix adaptation method in modern FL/LLM pipelines. In our experiments, across the considered setups, the sketched private variants consistently require strictly less added noise than GM-based baselines to reach the same privacy level, while maintaining competitive empirical performance. Thus, our contribution is twofold: a nontrivial theoretical extension of SGM to matrices together with its characterization, and an application of this matrix-valued mechanism in the LoRA-based federated learning settings.
>
> ---
>
> **Q2:** "Given the limited theoretical contribution, a strong empirical contribution is expected. However, this paper only evaluate the proposed method on CIFAR100 under a single privacy setting. The experiments only use ViT, while the paper motivates itself using LLMs. Moreover, the empirical results do not show advantage of the proposed method comparing to standard Gaussian mechanisms, in contrast to what the theory might suggest. "
>
> We thank the reviewer for this comment. When comparing the SGMM and SGMV versions of the algorithm with their non-sketched counterparts, whether the sketching versions can outperform is highly dependent on the choice of hyperparameters. According to Remark 2.1, when $b\geq\Omega\left(\frac{r\epsilon_{p}^{2}}{\ln\left(1/\delta_{p}\right)}\right)$, the noise needed by SGMM is smaller than that required by the classical GM. However, the sketching step can degrade the performance of the original algorithm, potentially counteracting the benefits from using less noise. Furthermore, from the comparison in Remark 2.1, the reduction in noise variance is more apparent when the sketching dimension $b$ (or, with a fixed compression rate, the original dimension $d$) is large, or when the privacy level $\epsilon_{p}$ is small.
>
> For example, we examine the setting where the privacy level is $\epsilon_{p}=0.75$, while keeping all other parameters, the model, and the dataset the same as in the experiments in the paper. In this case, the non-sketching version of private FFA-LoRA needs a noise multiplier of 3, SGMM-FFA-LoRA needs a noise multiplier of 2.0086, and SGMV-FFA-LoRA needs a noise multiplier of 1.0045, and we obtain the following table of accuracies. We can see that under this setting, the two sketching versions, SGMM-FFA-LoRA and SGMV-FFA-LoRA, both outperform the non-sketching version. We will add more kinds of experiments on different tasks and models in the final revision.
>
> | Algorithm                          | private FFA-LoRA (no sketching) | SGMM-FFA-LoRA | SGMV-FFA-LoRA |
> |------------------------------------|----------------------------------|---------------|---------------|
> | Accuracy                           | 0.4550                           | 0.4958        | 0.5256        |

---

> ### Author Response · Authors · 2025-12-01
> **Response to Reviewer YSJc**
>
> **Q3:** "Remark 2.1 states that "Comparing to classical GM, SGMM attains the same privacy level with smaller noise whenever the sketching dimension satisfies $b\geq\Omega\left(\frac{r\epsilon_{p}^{2}}{\ln\left(1/\delta_{p}\right)}\right)$." However, if we have $r=1$ and the matrix reduce to a vector, how the classical GM is better than the SGMM? I may misunderstand this point, and an explanation is appreciated."
>
> Remark 2.1 is obtained by comparing the lower bound on the variance of the Gaussian noise required by SGMM in Theorem 2.3, where $\sigma_{g}^{2}\geq\frac{c_{4}\sqrt{r}\tau^{2}\sqrt{\ln(1/\delta_{p})}}{\sqrt{b}\epsilon_{p}}$, with the corresponding bound for the classical Gaussian mechanism (GM), which is $\sigma_{g}^{2}\geq\frac{C\tau^{2}\ln\left(1.25/\delta_{p}\right)}{\epsilon_{p}^{2}}$. When the sketching dimension $b$ satisfies $b\geq\Omega\left(\frac{r\epsilon_{p}^{2}}{\ln(1/\delta_{p})}\right)$, the first bound is smaller than the second one, so SGMM requires less noise variance than GM in this regime.
>
> In the special case $r=1$, the matrix $\gamma(D)\in\mathbb{R}^{m\times r}$ reduces to a vector in $\mathbb{R}^{m}$, and SGMM specializes to the original Sketched Gaussian Mechanism (SGM) on vectors; in this case, Theorem 2.3 coincides with Theorem 2.1. Comparing Theorem 2.1 with the classical GM bound shows that when $b\geq\Omega\left(\frac{\epsilon_{p}^{2}}{\ln(1/\delta_{p})}\right)$, SGM (and hence SGMM with $r=1$) is preferable to GM because it requires smaller noise variance. Conversely, when $b\leq O\left(\frac{\epsilon_{p}^{2}}{\ln(1/\delta_{p})}\right)$, the classical GM bound can be tighter, and GM may require less noise than SGM. Thus, SGMM does not dominate GM uniformly for all sketching dimensions; it is advantageous precisely in the regime where the sketching dimension $b$ is sufficiently large relative to $\frac{\epsilon_{p}^{2}}{\ln(1/\delta_{p})}$ (or $\frac{r\epsilon_{p}^{2}}{\ln(1/\delta_{p})}$ in the general matrix case).
>
> This phenomenon is fully consistent with the original SGM analysis in Li et al. (2025): Remark 2.2 there explicitly states that, for vectors, SGM can outperform GM when the sketching dimension $b$ is relatively large, while GM can be better when $b$ is not large enough. Our statement in Remark 2.1 is therefore aligned with, and can be viewed as a matrix-valued generalization of, this earlier observation.
>
> ---
>
> **Minor:** "Minor issue in Line 110: citation Dwork et al. (2006) should use another format."
>
> Thank you for pointing this out. In the revised version, we will replace \cite with \citep for the citation to Dwork et al. (2006) to ensure the correct citation format.

---

### Meta-Review · Area_Chair_PZMG · 2026-01-03

**Summary:**

This paper extended Sketched Gaussian Mechanism to matrices for differentially private LoRA training in FL. This paper received a diverse set of review scores (2, 4, 4, 6). While reviewers appreciate the combination of theoretical and empirical studies, no reviewer championed this paper before and after discussion. Reviewers raised concerns on presentation clarity that leads to unclear contributions, weak empirical study on only one dataset (CIFAR-100) and one model, and concerns on shared sketch matrices. These concerns are not fully addressed in the authors' comments, and unfortunately reviewers did not actively engage in discussions.

I would encourage authors to incorporate the feedback to improve the paper: clarify contributions; for theory, clearly state the advantage over Li et al 2025, while considering comparing to [Privacy amplification via compression: Achieving the optimal privacy-accuracy-communication trade-off in distributed mean estimation](https://arxiv.org/abs/2304.01541) and  [Improved Communication-Privacy Trade-offs in L2 Mean Estimation under Streaming Differential Privacy](https://arxiv.org/abs/2405.02341); for experiments, improve on models and datasets.

**Reviewer Concerns:**

Reviewers raised concerns on presentation clarity that leads to unclear contributions, weak empirical study on only one dataset (CIFAR-100) and one model, and concerns on shared sketch matrices. These concerns are not fully addressed in the authors' comments, and unfortunately reviewers did not actively engage in discussions.

**Reviewer Scores:**

This paper received a diverse set of review scores (2, 4, 4, 6). The reviewers did not actively engage in discussions. The authors' rebuttal on clarifying impact from empirical results may be useful for an average score increase of 1-2.

---

### Decision · Program_Chairs · 2026-01-26

Reject